# Larger rock extraction sites could improve the efficiency of enhanced rock weathering in the United Kingdom
M. Madankan[1], E. P. Kantzas [2], R.M.E. Espinosa [3], S. H. Vetter [4], L. Koh[3], P. Smith [4], D. J. Beerling [2] & P. Renforth [1]✉

Large-scale removal of carbon dioxide from the atmosphere is required to meet net-zero targets. Enhanced rock weathering, in which crushed silicate minerals are spread on cropland soils, is a promising approach, but the logistics of its supply chain are poorly understood. Here, we use a numerical spatio-temporal allocation model that links potential rock extraction sites in the United Kingdom with croplands, modelling deployment pathways over the period 2025–2070. We find that expanding individual quarries (up to 20 times larger than the current average) and prioritising supply timing and location can increase carbon-removal efficiency by 20%, cut transport demand by 60% and reduce the number of operating quarries four-fold, while enabling up to 700 million tonnes of carbon dioxide removal by 2070. However, these large sites may face stronger local opposition and planning challenges, underscoring the critical role of policy in enabling feasible deployment.

The urgency of addressing climate change has intensified as the global mean temperature already breached the 1.5 °C threshold in 2024, averaging about 1.55 °C above the pre-industrial baseline[1]. Recent IPCC findings[2,3] stress that returning below this level will require large-scale carbon dioxide removal (CDR) in addition to deep emissions cuts, so that net-negative $CO_2$ balances can be achieved. Integrating CDR strategies into a broader portfolio of climate mitigation solutions is crucial to achieve net-zero emissions by mid-century[4].

Enhanced rock weathering (ERW), a promising geochemical CDR method[5], involves amending soils with crushed calcium- and magnesium-rich silicate rocks and has shown substantial potential to sequester $CO_2$ at scale[6–9]. The scalability of ERW, crucial for achieving net-zero targets, requires the sustainable and efficient supply of silicate rock. While research has recently focused on the chemical and agronomic impacts[8,10,11], environmental controls[12,13] and quantifying $CO_2$ removal of ERW[14], the upstream logistics of rock extraction and supply chain management remain largely unexplored.

Life-cycle studies show that the logistics and the electricity mix are key controlling parameters of the effectiveness of CDR, especially through ERW[9,15–17]. Road and rail transport footprints caused by long quarry to field distances (in the USA and India for instance) account for 20–40% of every midpoint impact, while grinding energy adds 40–80%[15]. Moving from a

business-as-usual energy mix to a 2 °C scenario energy grid can cut the $CO_2$ footprints of these processes by 55% and lift net-removal efficiency[15]. Beyond energy and transport, effective source-to-field allocation of feedstocks needs to be considered as part of the ERW supply chain since weathering yields vary spatially across croplands[18] and directly impact the effectiveness of CDR through ERW. This work extends previous studies by exploring how spatial and temporal variations in upscaling and source-to-field allocation within the ERW supply chain control its effectiveness.

The UK, with extensive managed croplands and abundant basic silicate rock deposits, is particularly well-suited for large-scale deployment of ERW[8,18–20]. Modelling studies estimate that ERW could achieve a net CDR of up to 30 $MtCO_2$ $yr^{-1}$ in the UK by 2050, contingent on the supply of over 168 Mt $yr^{-1}$ of basic silicate rock annually[18]. However, a recent inventory of UK mineral resources for ERW has identified current basic silicate rock production at approximately 15 Mt $yr^{-1}$, with a total permitted reserve of around 0.5 Gt of rock[20]. Achieving the target of 30 $MtCO_2$ $yr^{-1}$ of net CDR will thus require more than a tenfold increase in current basic silicate rock extraction by 2050.

Such a substantial expansion within a relatively short timeframe, while not unprecedented (for example, a rise in the UK's crushed-rock production from ~25 Mt $yr^{-1}$ in 1945 to over 200 Mt $yr^{-1}$ by 1990[18]), necessitates a robust plan for the spatial distribution and temporal deployment of this

[1]Research Centre for Carbon Solutions, School of Engineering and Physical Sciences, Heriot-Watt University, Edinburgh, UK. [2]Leverhulme Centre for Climate Change Mitigation, School of Biosciences, University of Sheffield, Sheffield, UK. [3]Advanced Resource Efficiency Centre, Management School, University of Sheffield, Sheffield, UK. [4]Institute of Biological and Environmental Sciences, School of Biological Sciences, University of Aberdeen, Aberdeen, UK. ✉e-mail: P.Renforth@hw.ac.uk

                    1

**Table 1 | Summary of scenario codes, descriptive names and key differentiating features: target supply by 2070 (Mt yr$^{-1}$), per-quarry capacity cap (Mt yr$^{-1}$) and contributing quarry types**

| Scenario name | Descriptive name | Target supply by 2070 (Mt yr$^{-1}$) | Max cap per quarry (Mt yr$^{-1}$) | Contributing quarry types |
|---|---|---|---|---|
| S1 | Low rock supply | 32 | 1 | Active only |
| S2.a | Medium rock supply—Active only | 97 | 2 | Active only |
| S2.b | Medium rock supply—Active and inactive | 97 | 1 | Active + inactive |
| S3.a | High rock supply—Active only | 166 | 5 | Active only |
| S3.b | High rock supply—Active and inactive | 166 | 2 | Active + inactive |
| S3.c | High rock supply—Active, inactive and new | 166 | 1 | Active + inactive + new |

upscaling to ensure a sustainable increase in rock supply. The technical feasibility of this plan will be shaped by rock geochemistry as a controlling factor for the maximum CDR potential of rocks, current production capacity of existing quarries, potential expansion rates, the possibility of de-mothballing inactive quarries, opening new extraction sites and transport distances between croplands and rock sources. Upstream rock supply processes not only play a crucial role in determining the CDR potential and cost of ERW, but also substantially influence its environmental and social impacts[15]. Moreover, public support for ERW is shaped by how the technology is implemented, particularly concerning environmental impacts, the scale of deployment, mining practices and the transport of rock materials[21],[22]. Although a full appraisal of social and environmental impacts of expanded rock extraction demands comprehensive, site-specific studies, an initial indicator of their potential extent can still offer high-level insight for comparing up-scaling scenarios and identifying where detailed assessment should be prioritised.

Using the UK as a case study, this work aims to develop a comprehensive model for the spatial and temporal allocation of expanded rock production capacity. We analyse three rock supply scenarios, each with sub-scenarios based on strategies for scaling up production. These strategies include the expansion of active quarries, reactivation of inactive quarries and the opening of new quarries. By integrating geochemical data, transport logistics, production capacities and some environmental and social constraints, our model aims to identify the most efficient pathways for upscaling rock supply, thereby maximising the net CDR achievable through ERW in the UK.

## Results

We simulated rock extraction under three rock supply scenarios, each with different production targets by 2070: S1, low rock supply (32 Mt yr$^{-1}$), S2, medium rock supply (97 Mt yr$^{-1}$) and S3, high rock supply (166 Mt yr$^{-1}$). These scenarios, which are consistent with scenarios defined in previous studies[18], necessitate substantial upscaling. This upscaling is modelled as an annual growth rate applied to the existing capacity of individual quarries until their production reaches a specified maximum annual cap. Our initial analysis assessed the impact of varying annual growth rates and production caps on the upscaling pathways and resulting CDR. Historical data on crushed rock production from 1960[18] to the present reveal that annual growth rates can vary widely, reaching up to 50%. These fluctuations are driven by variable supply and demand, but are not necessarily indicative of the potential rate; the largest 10-year rolling average of 15% may be a more suitable indicator of potential rate. Consequently, we constrained scenarios with a base-case of 15% but examined the model sensitivity of growth rates ranging from 5 to 50%, and considered three annual production caps of 1, 2 and 5 Mt yr$^{-1}$ for individual quarries. A summary of the scenarios simulated in this study, along with their features, is presented in Table 1.

Across all scenarios with the same cumulative rock extracted (See Supplementary Information Fig. S1), more rapid annual growth rates in quarry capacity led to larger cumulative CDR. This effect is particularly substantial when growth rates increase from 5 to 15%, potentially resulting in up to 15% more CDR. Similarly, increasing the annual production cap

from 2 to 5 Mt yr$^{-1}$ per quarry led to an additional 13% in CDR by 2070. Lifting constraints on growth rates and production caps enables the model to prioritise quarries with more favourable conditions, allowing them to contribute more substantially and earlier to the rock supply, thereby enhancing CDR efficiency.

Our models showed that relying only on the expansion of existing active quarries for fulfilling the target demand requires over 30 mega-quarries (quarries with over 1 Mt yr$^{-1}$ capacity) to operate by 2070 (Supplementary Information Fig. S2). To fulfil the target demand, the pathways were designed by following one or a combination of two or three strategies: expansion of currently active quarries, reactivating inactive quarries and opening new quarries (Supplementary Information Fig. S3).

### Cumulative rock supply

Deployment scenarios have been compared by metrics including cumulative rock extracted (between 2025 and 2070), cumulative CDR, transport capacity required, average CDR rate and the number of unique quarries contributing to rock supply for all scenarios (S1, S2.a, S2.b, S3.a, S3.b and S3.c). The results are summarised in Table 2.

In Scenario S1 (low rock supply), a cumulative total of 1.1 Gt of rock is extracted, reflecting a conservative approach focused on expanding currently active quarries. This results in a cumulative CDR of 175 Mt CO$_2$. In Scenarios S2.a (Medium rock supply, active only) and S2.b (Medium rock supply, active and inactive), the cumulative rock extracted is substantially greater, at 2.4 Gt, driven by moderate upscaling efforts. Scenario S2.a achieves a cumulative CDR of 348 Mt CO$_2$ by 2070, due to higher production caps at active quarries. In contrast, Scenario S2.b, which includes reactivating inactive quarries, achieves a slightly lower cumulative CDR of 336 Mt CO$_2$ by 2070 due to the limited production cap of 1 Mt yr$^{-1}$. Please note that the efficiency drop from S2.a to S2.b is not inherently due to adding inactive quarries. Higher production caps (e.g. 2 Mt yr$^{-1}$) give the model more freedom to allocate production to the most favourable sites, whereas a tighter 1 Mt yr$^{-1}$ cap spreads production across a wider, and generally less optimal, pool of quarries.

The high rock supply scenarios (S3.a,b,c) involve the larger extraction of 4.7 Gt of rock, reflecting a substantial upscaling effort. Scenario S3.a (High rock supply, active only) achieves a cumulative CDR of 703 Mt CO$_2$, demonstrating the efficiency of concentrating rock extraction in a smaller number of high-capacity quarries (47 mega quarries). Scenario S3.c, (High rock supply, active, inactive and new), on the other hand, achieves a lower cumulative CDR of 58 Mt CO$_2$ and faces higher logistical demands (1975 t.km tCO$_2^{-1}$) compared to S3.a (1227 t.km tCO$_2^{-1}$). Scenario S3.b (High rock supply, active and inactive), which combines the expansion of active quarries with the reactivation of inactive quarries, achieves a cumulative CDR of 639 Mt CO$_2$, indicating the benefits of a balanced approach.

The mean CDR rate (t CO$_2$ per t rock) declines as supply targets increase, illustrating a clear scale-efficiency trade-off. Because the model always allocates rock first to the highest CDR yield quarry-cropland pairs, average CDR rate (efficiency) inevitably falls as supply targets rise: once the allocations of most favourable quarry-cropland pairs have been fulfilled, meeting the larger medium and high supply targets (S2 and S3) requires the model to draw on progressively less favourable combinations. Lower

**Table 2 | Cumulative metrics (for the years ranging 2025–2070) including cumulative rock extracted, cumulative CDR, transport capacity required, average CDR rate and the number of unique quarries contributing to rock supply for all scenarios (S1, S2.a, S2.b, S3.a, S3.b and S3.c)**

| Scenario | Cumulative rock extracted (Gt) | Cumulative CDR (Mt) | Transport (t.km $tCO_2^{-1}$) | Average road transport distance (km) | Mean CDR rate ($tCO_2$t $Rock^{-1}$) | Number of supplying quarries (count) |
|---|---|---|---|---|---|---|
| S1 | 1.07 | 183 | 1057 | 178 | 0.17 | 46 |
| S2.a | 2.39 | 377 | 1438 | 224 | 0.16 | 63 |
| S2.b | 2.39 | 364 | 1377 | 208 | 0.15 | 114 |
| S3.a | 4.71 | 758 | 1140 | 183 | 0.16 | 47 |
| S3.b | 4.71 | 693 | 1539 | 226 | 0.15 | 90 |
| S3.c | 4.71 | 634 | 1822 | 246 | 0.13 | 183 |

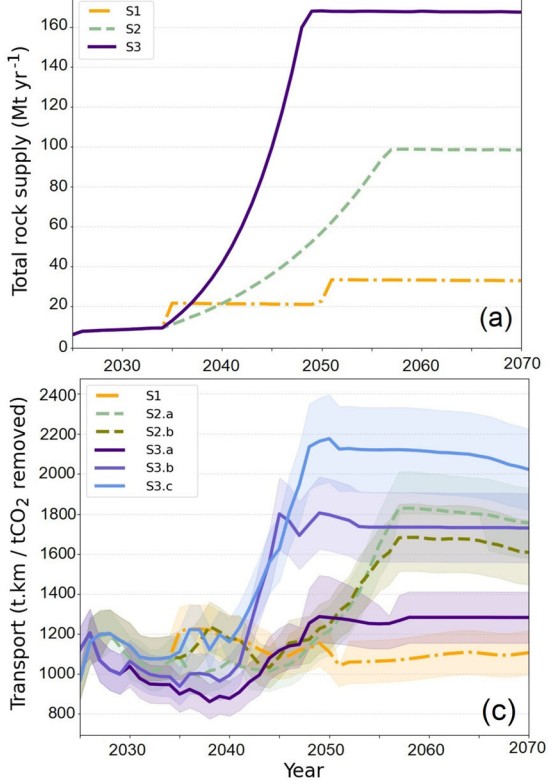

**Fig. 1 | Potential of enhanced rock weathering in the UK 2025–2070. a** Target annual rock supply scenarios from Kantzas et al. (2022), **b** corresponding annual net carbon dioxide removal, **c** transport requirements, and mean CDR **d** efficiency per tonne of rock. Description of scenarios: S1, Low rock supply scenario, cap 1 Mt yr$^{-1}$. S2.a, Medium rock supply scenario, cap 2 Mt yr$^{-1}$. S2.b, Medium rock supply scenario, cap 1 Mt yr$^{-1}$. S3.a, High rock supply scenario, cap 5 Mt yr$^{-1}$. S3.b, High rock supply scenario, cap 2 Mt yr$^{-1}$. S3.c, High rock supply scenario, cap 1 Mt yr$^{-1}$. Shaded envelopes represent the uncertainty arising from variation in rock max CDR potential. (see Supplementary Data 1).

per-quarry annual production caps accentuate this effect by spreading production across more, and often less favourable quarries, whereas higher caps (e.g. 2 and 5 Mt yr$^{-1}$ in S2.a and S3.a, respectively) let extraction concentrate at the most efficient sites and partly offset the decline.

Rock transport required per tonne of $CO_2$ removed from the atmosphere has direct implications for the CDR efficiency. Scenario S2.a (Medium rock supply, active only) demonstrates the lowest transport requirements among all scenarios, while S3.a (High rock supply, active only) shows the best efficiency among the high supply scenarios. The average CDR rate provides insight into the effectiveness of $CO_2$ removal per unit mass of rock used and has implications for the CDR cost associated with rock extraction, grinding, transport and spreading, with scenario S1 (low rock supply) and S3.c (High rock supply, active, inactive and new) having the highest and lowest rate respectively.

### Temporal evolution of rock supply

Target rock demand scenarios along with corresponding temporal evolution of key metrics, including CDR, transport capacity required and CDR rate for all scenarios is presented in Fig. 1.

The annual CDR, depicted in Fig. 1.b, shows an increasing trend across all scenarios over time consistent with the rock supply. By 2070, the CDR can range from 6 to 27 Mt yr$^{-1}$ of $CO_2$, depending on the scenario chosen. Despite having the same amount of rock supplied, the CDR achieved varies substantially between sub-scenarios within S3 (High rock supply). For instance, Scenario S3.a (High rock supply, active only) shows (~20%) higher CDR compared to S3.c (High rock supply, active, inactive and new), and a more pronounced difference (~35%) in transport requirement. However, prior to 2045, these differences in CDR between sub-scenarios are minimal due to the rapid expansion required, necessitating the utilisation of all

available resources and leaving little room for optimisation. But, after 2050, as demand stabilises and quarry capacities have expanded, the model can better optimise resource allocation, resulting in substantial variations in CDR among different sub-scenarios. By 2070, the S3.a (High rock supply, active only) captures up to 5 Mt yr$^{-1}$ more $CO_2$ than S3.c (High rock supply, active, inactive and new), highlighting the effectiveness of optimised resource allocation.

From 2025 to 2035, the transport requirements decrease while the CDR rate increases. This period reflects the model's ability to prioritise effectively under relatively low demand. However, between 2035 and 2050, the trend reverses: transport requirements increase, and CDR rates decrease due to the need to supply croplands at greater distances and the rapid surge in demand, which limits the model's optimisation potential. Post-2050, as demand stabilises, the trend improves again and the difference in efficiency between sub-scenarios (especially within high rock supply scenarios, S3) becomes more pronounced, showcasing the enhanced optimisation potential under stable demand and up-scaled rock supply conditions.

Among high supply scenarios, Scenario S3.a (High rock supply, active only) has the most efficient logistics over time, followed closely by S3.b (High rock supply, active and inactive), while S3.c (High rock supply, active, inactive and new) faces the highest transport requirements. Post-2050, scenario S3.a exhibits a more pronounced decline in transport requirements compared to other scenarios, so that by 2070, S3.a achieves approximately 35% less transport effort per unit mass of CDR than S3.c. Scenario S3.c exhibits the lowest average CDR rate among all scenarios, whereas S3.a appears more efficient among the high rock supply scenarios. Notably, after 2055, S3.a surpasses the efficiency of even the S2 (Medium rock supply) scenarios.

## Spatial distribution of rock extraction for ERW

The spatial and temporal distribution of rock extraction sites for ERW in the UK by 2070 for the largest rock supply scenario (S3) is shown in Fig. 2. The varying combination of active, inactive and new quarries results in gradual increase in the number and geographic reach of quarries by moving from scenario S3.a (High rock supply, active only) to S3.b (High rock supply, active and inactive) and S3.c (High rock supply, active, inactive and new), leading to a broader distribution across the UK in S3.c. However, the overall pattern remains uneven, with extraction sites predominantly concentrated in the central belt of Scotland and Northern Ireland (NI).

In Scenario S3.a, which relies solely on the expansion of currently active quarries, rock supply is met by a smaller number of quarries, with gradual increases in their production capacity and high-capacity quarries (up to 5 Mt yr$^{-1}$) emerging from 2050. Notably, in 2040, the number of contributing quarries in NI temporarily decreases, highlighting the model's preference for utilising a minimum number of more favourable quarries.

Scenario S3.b involves a reactivation of inactive quarries from 2040, with a combination of existing and reactivated quarries contributing to production toward 2070. Several quarries are reactivated around the Midlands and North-East England with close proximity to abundant croplands.

From 2050 and continuing to 2070 in Scenario S3.c, new quarries begin operations and contribute to the rock supply alongside reactivated and existing quarries. Due to the maximum cap for each individual quarry being limited to 1 Mt yr$^{-1}$, reactivated quarries, such as those in NI, start contributing to the rock supply as early as 2040 to ensure fulfilling the target demand. The distribution of new quarries is concentrated mainly in the Central Belt of Scotland, NI, Northern Wales, South-West and North-East England and the North West of Scotland.

The background map in Fig. 2 shows the cropland grid cells and the amount of rock spread onto cropland in each grid cell. While croplands nearer to quarries receive rock in the early years, most of the supplied rock is ultimately applied to the central and eastern regions of England. This is because these regions have more extensive areas of cropland. Additionally, a substantial portion of the rock supply from NI reaches these regions via the

Felixstowe port, owing to its proximity to high-efficiency croplands and convenient transport logistics. See Supplementary Information Fig. S4 for the locations of the main ports and the shipping routes contributing to the supply of rock from NI to Great Britain (GB) for ERW.

In addition to spatial distribution, the temporal prioritisation in the model has led to varying contributions from active, inactive and new quarries to the total rock supply for ERW over time, as shown in Fig. 3. In Scenarios S3.b (High rock supply, active and inactive) and S3.c (High rock supply, active, inactive and new), the contribution to the total rock supply exhibit substantial shifts over the years. Initially, active quarries dominate the supply, but as time progresses, reactivated and new quarries increasingly contribute, especially post-2045. In scenario S3.c, by 2070, new quarries form a substantial part of the supply chain, highlighting the model's strategy to open new extraction sites to meet the increasing demand. After 2050, as the demand stabilises, existing quarries have had sufficient time to expand their production capacity. Consequently, the model allocates more rock to expanded existing quarries, leading to a slight decline in the contribution from new and reactivated quarries toward 2070.

## Resource requirements, flow and logistics

The analysis of rock supply requirements by counties and countries for ERW in the UK reveals substantial regional variation in the demands and reserves needed by 2070. Fig. 4 illustrates these differences across the three largest rock supply scenarios (S3.a, S3.b and S3.c).

In 2070, Scenario S3.a (High rock supply, active only) shows a substantial concentration of rock extraction activities in smaller number of counties, with capacities exceeding 25 Mt yr$^{-1}$ in some jurisdictions. Scenario S3.b (High rock supply, active and inactive) introduces notable contributions from North Scotland, Midlands and North East England in addition to the contributing counties in S3.a. Scenario S3.c (High rock supply, active, inactive and new), with its lowest production cap per individual quarry (1 Mt yr$^{-1}$), results in a more widespread but less intensive distribution of extraction activities across counties.

The total reserve requirement by 2070 also highlights these regional differences. In S3.a (High rock supply, active only) and S3.b (High rock supply, active and inactive), NI and the Central Belt of Scotland require substantial reserves, up to 800 Mt in some counties. Scenario S3.c (High rock supply, active, inactive and new) distributes the reserve requirements more evenly across the UK, though NI and Scotland still dominate.

The concentration of rock extraction activities within a specific area is illustrated by county in Fig. 4 (bottom row). The data reveal a high concentration of extraction activities in NI (particularly Northern Antrim and Derry/Londonderry) and the Central Belt of Scotland (Renfrewshire, North Lanarkshire, Edinburgh). Progressing from Scenario S3.a (High rock supply, active only) to S3.b (High rock supply, active and inactive) and S3.c (High rock supply, active, inactive and new), the intensity of these extraction activities is distributed more evenly across additional counties, reducing the concentration in any single county and thus potentially reducing the localised environmental and social impacts.

County-based extraction density maps can distort the true burden of quarrying activities because counties vary widely in size: high extraction in a large county may appear as a low extraction density, while the same extraction in a small county yields a much higher extraction density. This area dependence can obscure real intensity peaks. However, county boundaries correspond to local-authority administrative regions responsible for planning permissions and community impacts, making county-scale density maps directly applicable for policymakers. For a more consistent spatial view, we also computed extraction density on a uniform $10 \times 10$ km grid (Fig. S6 in Supplementary Information). While these grid-based patterns generally align with the county-level results, they more clearly show how higher per-quarry production caps (S3.a) concentrate high-density extraction into a few hotspots, whereas more constrained production caps (S3.b and S3.c) diffuse this burden more evenly across the UK.

The contribution of the UK's countries to rock extraction and spreading on croplands by 2070 is shown in Fig. 5a, for Scenario S3.c (High

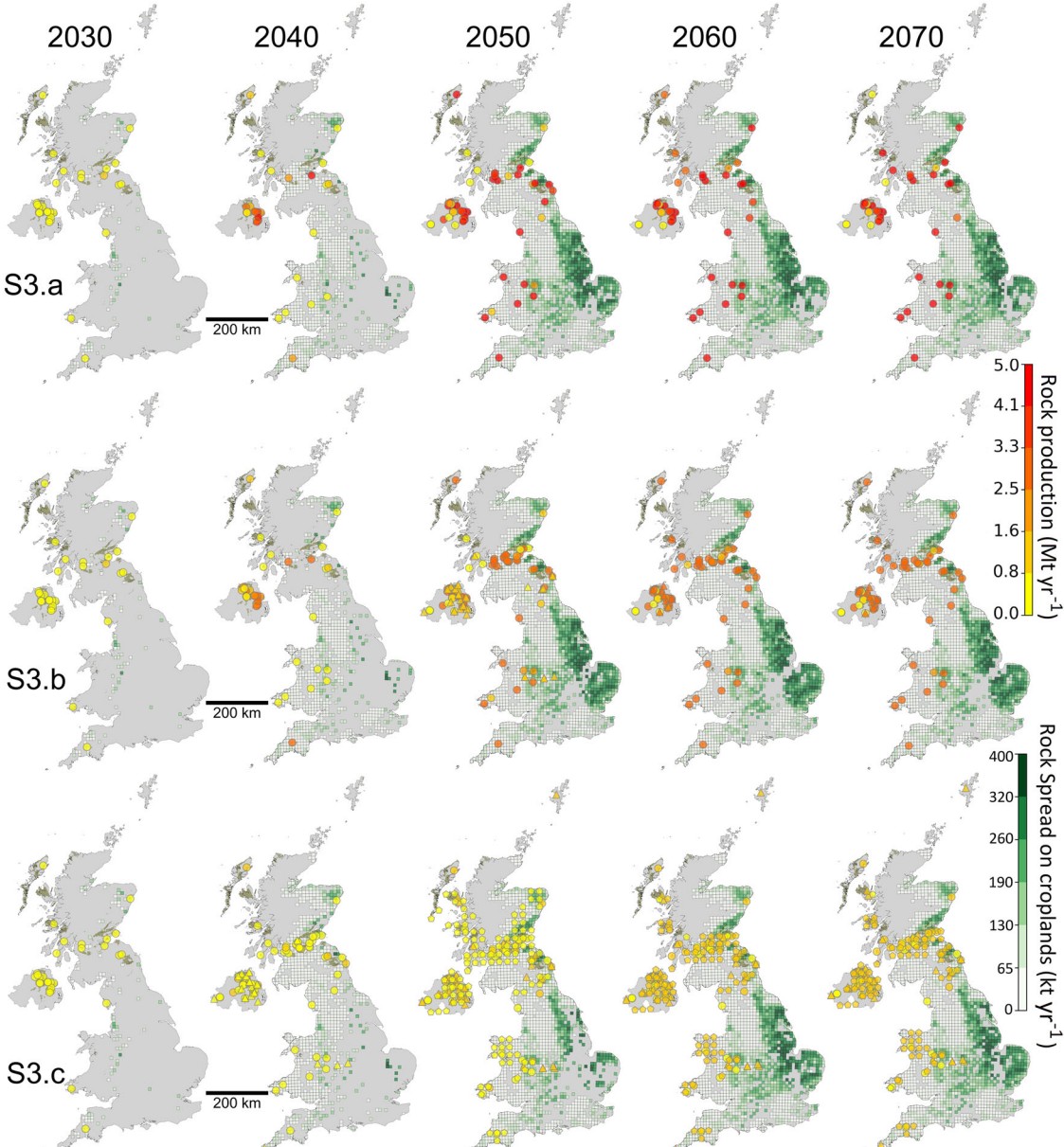

**Fig. 2 | Spatial and temporal distribution of quarries and their annual production capacity.** Maps show annual rock production from quarries and the quantity of rock spread on croplands ($10 \times 10$ km grid cells) across the UK for years 2030, 2040, 2050, 2060 and 2070 under three sub-scenarios: S3.a (top row), S3.b (middle row) and S3.c (bottom row). The background grid represents rock application on cropland, shaded from white to dark green, where darker shades indicate greater quantities of rock spread per grid cell.

rock supply, active, inactive and new). The flow of rock supply from producing countries to croplands of different countries where the rocks will be spread for ERW is shown by flow channels with width of each flow is proportional to the quantity of rock (in million tonnes) transported.

NI and Scotland are the primary rock producers, contributing a combined total of 123.3 Mt yr$^{-1}$, which accounts for over 70% of the total rock supply. England, which produces 27.4 Mt yr$^{-1}$ (16% of the total rock supply), is the main region for rock application, with 141.6 Mt yr$^{-1}$ being spread on its croplands by 2070, representing more than 80% of the total rock spread. NI, the largest producer with 66.7 Mt yr$^{-1}$ transports all extracted tock to England and Scotland. This indicates a substantial need for dedicated transport routes.

Figure 5b, cpresents the UK's historical domestic freight transport by mode (road, rail, water) alongside the additional capacity required for ERW rock haulage through to 2070. From 1953 to about 2000, total freight rose

from roughly 90 Gt·km to a peak near 225 Gt·km, then declined to around 155 Gt·km by 2020. We project baseline freight demand under two compound-growth projections, 1.1% p.a. and 0.7% p.a[23]. (NIC, 2019), shown in Fig. 5b, c, respectively. Onto each baseline we add ERW demand for the most transport-intensive scenario (S3.c), which begins around 2025, ramps up to ~50 Gt·km yr$^{-1}$ by 2045 and then plateaus. By 2050, the sum of baseline plus ERW freight demands surpasses the historic peak in both projections, reaching approximately 330 Gt·km by 2070 under 1.1% growth and 270 Gt·km under 0.7% growth.

## Discussion

This study presents a comprehensive framework for assessing and planning the rock supply chain necessary for national-scale deployment of ERW at scale. The trade-offs between maximising CDR potential, managing transport logistics, ensuring practical feasibility and addressing social and

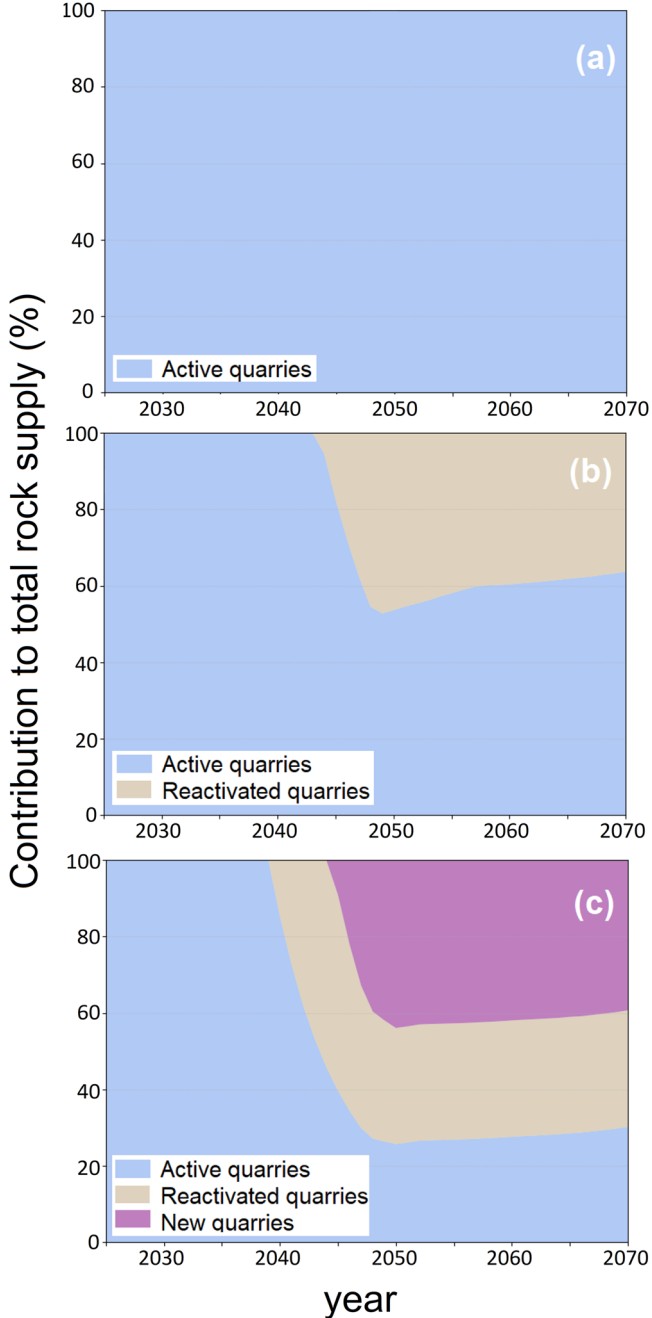

**Fig. 3 | Normalised contribution of active, reactivated and new quarries to the rock supply for ERW in the UK. a** Scenario S3.a, (High rock supply, active only), **b** Scenario S3.b, (High rock supply, active and inactive) and **c** Scenario S3.c (High rock supply, active, inactive and new). (see Supplementary Data 2).

environmental impacts are crucial for the effective deployment of ERW. Our findings highlight the framework's critical role in developing effective ERW deployment pathways for national policymakers aiming to upscale ERW as part of their CDR efforts.

The scenarios for rock supply up-scaling show varying levels of CDR potential from 4 to 25 Mt yr$^{-1}$ by 2050, depending on the approach to quarry expansion, which is comparable with CDR projections of 6–30 Mt yr$^{-1}$ by 2050 from other study[18]. This could make a substantial contribution to the expected 100 Mt yr$^{-1}$ CDR required in the UK by 2050[24]. Expanding fewer quarries to high capacities allows for optimisation, by focusing on quarries with the best conditions, such as proximity to crop-lands and high-quality rock, the model can maximise CDR efficiency. In the

scenario with fewer but larger quarries (S3.a, High rock supply, active only), allowing quarries with favourable conditions to contribute more to the rock supply, enhances CDR efficiency. So, the higher CDR efficiency offsets for the extra transport distance, reducing the overall transport footprint per unit $CO_2$ removed.

In our scenarios, average rock transport distances of 183 km (S3.a), 226 km (S3.b) and 246 km (S3.c) exceed the UK's typical aggregate haul of ~45 km (28 miles)[25]. Although larger aggregate transport distances have been already occurred in other countries (e.g. 111 km maximum in Spain) as quarries are sited farther from demand centres[26]. As Kantzas et al. (2022)[18] demonstrate, transport remains a minor component of total ERW cost in their UK study. While extended transport distances inevitably raise both costs and environmental footprint, the net carbon removal benefits outweigh the additional transport-related emissions, and the premium value of $CO_2$ removal credits is expected to offset the higher logistics expenses.

Country-level material flow breakdown reveals that NI, as a primary supplier of rock for ERW by 2070, will need to export over 60 Mt of rock annually to England, Scotland and Wales. Presently, 11.5 Mt of goods are exported through NI's ports, up from 5.5 Mt in 1998[27]. Additionally, according to the recent Aggregates Mineral Survey[28], England and Wales imported approximately 4.8 Mt of aggregates from Scotland and Europe. Given this context, a substantial increase to over 60 Mt annually necessitates future investment in shipping routes and port infrastructure to handle the increased volume efficiently. For example, a 6 Mt capacity enhancement at the Port of Melbourne costs approximately $1.5 billion[29]. Applying benchmarks from bulk material ports, the capital expenditure for a 60 Mt yr$^{-1}$ capacity port in NI can be in the scale of tens billion dollars. Over a 50-year service life, the breakdown of this capital expenditure per tonne of rock transported through this port can be approximately $20 t$^{-1}$. While this might be considered as high upfront capital expenditure, maritime shipping's operational costs and $CO_2$ footprint are about an order of magnitude lower than those of road haulage[19], reinforcing the economic and environmental case for leveraging sea routes in the long-haul segment of the ERW supply chain.

Up-scaling rock extraction for ERW could lead to environmental and social impacts that are rarely quantified[15]. The relative social and environmental impact of individual basic silicate rock quarries has been considered in our previous work[20], as an approximate baseline indicative of the possibility of planning approval. Here, we quantify extraction density which may be an approximate indicator for the environmental and social impacts of up-scaled rock extraction. Measuring the cumulative amount of rock extracted from a unit area allows for a better understanding of the intensity of potential impacts in specific regions.

Balancing local environmental impacts with the broader economic benefits of resource extraction is a long-standing challenge, in which local communities often bear the impacts from quarrying, but the economic benefits are dispersed more widely[30]. This is apparent in these modelled scenarios in which 7% of UK local authorities would be responsible for over two-thirds of total rock production (as in Scenario S3.a, High rock supply, active only), yet the potential agricultural benefits would be concentrated in the east of England, and the CDR removal benefits would be national.

Land use change can be one of the key environmental impacts of expanded rock extraction[31] (see Supplementary Text 1). By overlaying our projected quarry expansions onto the UK Land Cover Map (Supplementary Information Fig. S5a), we estimate that scenario S3.a (High rock supply, active only), as most ambitious scenario, would convert roughly 33,000 ha of land by 2070, primarily improved grassland, deciduous woodland and arable lands, yet the area of each single affected land use type remains below 0.1% of its national extent (Supplementary Information Fig. S5b). However, based on our extraction-density analysis (Fig. 4), these small land use changes can be heavily clustered in particular regions, where their negative impact may be more pronounced and thus requires detailed, site-level assessment and targeted mitigation.

**Fig. 4 | Annual rock extraction in 2070, cumulative reserve requirements and Extraction density across UK counties under S3 sub-scenarios.**
**a** Annual rock production by county in year 2070
**b** cumulative reserve required by 2070 by county in different S3 (High rock supply) sub-scenarios.
**c** Extraction density as a metric representing the cumulative reserve requirement divided by the county area, illustrating the potential concentrated environmental and social impact of extraction activities in each county.

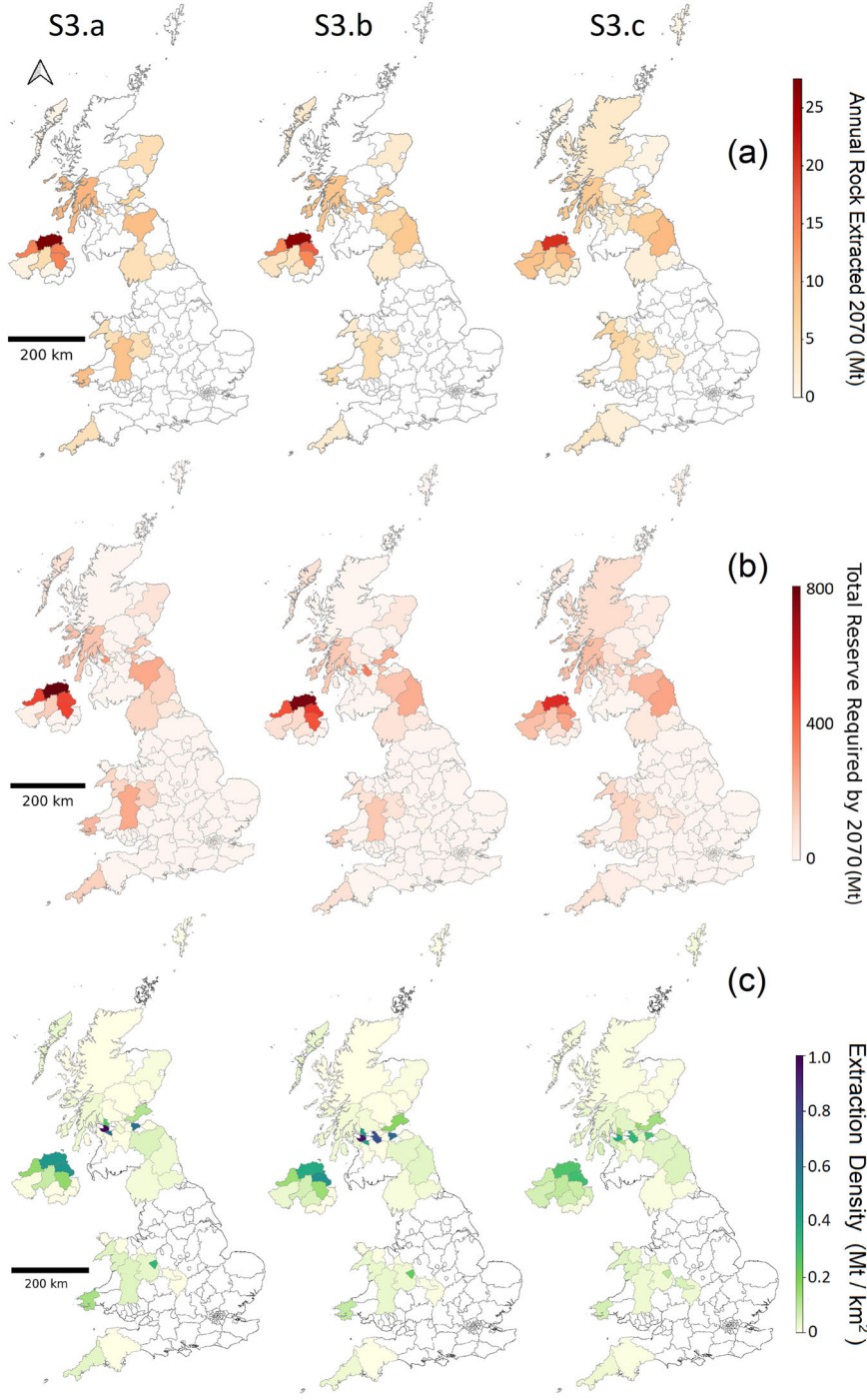

Biodiversity is another key environmental factor that can be affected by quarrying. Reviewing the relationship between quarrying and biodiversity in the UK reveals a complex interplay of impacts and opportunities, with studies highlighting both the detrimental effects of habitat removal, fragmentation, dust, noise and hydrological change[32,33] and the potential for substantial ecological benefits when restoration is planned from the outset. Immediate losses in species richness and ecosystem function can be mitigated, and even reversed through robust environmental impact assessments, detailed biodiversity action plans and innovative restoration techniques that recreate wetlands, woodlands and species-rich grasslands diverse native fauna and flora[34–37].

In addition to upstream supply-chain impacts, potential downstream effects on soils also warrant consideration. These include, but are not limited to, trace-metal accumulation (e.g. Ni, Cr, Cu) from prolonged silicate

addition, reductions in soil porosity and hydraulic conductivity due to fine mineral dust and soil-pH shifts that may limit micronutrient availability and disrupt microbial communities[38,39]. Detailed, field-specific assessments, incorporating feedstock geochemistry and local soil parameters, will be essential to integrate these geochemical risk metrics alongside CDR optimisation in future deployment pathways.

While detailed policy design is beyond the scope of this study, mechanisms such as channelling a portion of ERW carbon-credit revenues back into affected regions for local restoration funds or embedding restoration and monitoring obligations into quarry permitting could help ensure that communities hosting the most intensive extraction both benefit economically and see the negative impacts mitigated.

Up-scaling of ERW feedstock supply, will require planning permission for new rock extraction and establishing necessary local infrastructure.

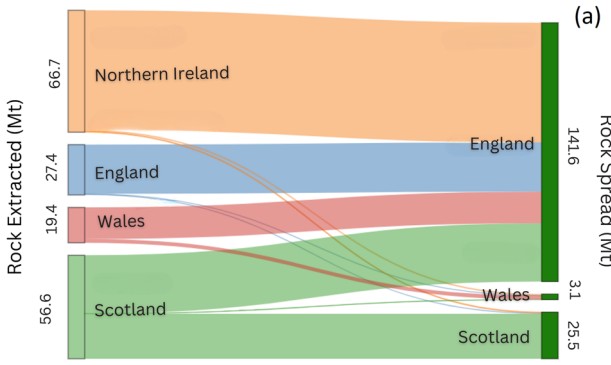

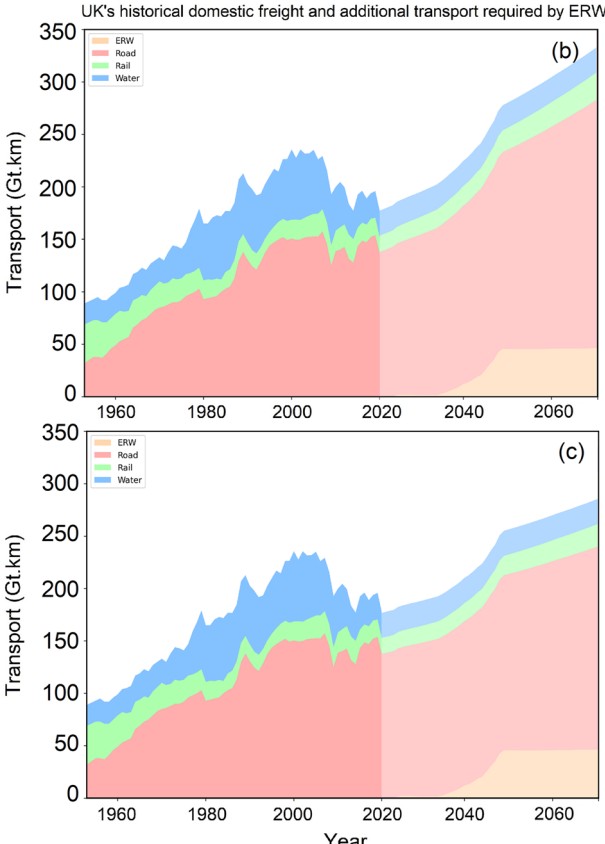

**Fig. 5 | Transport requirement for ERW in the UK under the high rock supply scenario (S3.c). a** Flow of rock supply from producing countries to croplands of different countries where the rocks will be spread for ERW. Width of each flow channel is proportional to the quantity of rock transported. And historical domestic freight transport in the UK and projected additional transport capacity required by Enhanced Weathering (ERW) by 2070, assuming 1.1% (**b**) and 0.7% (**c**) annual growth rate for baseline freight demand. These results are based on the data of transport requirement for scenario S3.c (High rock supply, active, inactive and new) as the most transport-intensive scenario. (see Supplementary Data 3).

Under the most ambitious scenario (S3, High rock supply), NI, as the principal source of rock for ERW, would supply roughly 60 Mt yr$^{-1}$ of basic silicate rock by 2070. This output is 24 times its current basic silicate rock production (~2.5 Mt yr$^{-1}$) and five times its total aggregate production (~12 Mt yr$^{-1}$; DfE, 2023). Nationally, the UK would require about 150 Mt yr$^{-1}$ of rock for ERW in 2070, a volume equal to the total current aggregate consumption of England and Wales[28].

It should be noted that in our model, ERW feedstock is treated as an additional allocation on top of today's aggregate demand. Future rise in aggregate demand derived from housing or infrastructure targets could

create competition for rocks. On the other hand, the incremental contribution of recycled and secondary aggregate, which already accounted for around 28% (69.6 Mt) of UK aggregates in 2021[40], is expected to grow under circular-economy initiatives. This expansion of recycled supply can alleviate pressure on primary extraction, helping to ensure that both construction aggregate and ERW feedstock requirements to be met without conflict.

Based on the Mineral Products Association report[41], permitted reserves for crushed rock are currently being replenished at a slower rate than extraction, creating a reserve gap. Although our model is founded on the UK's abundant basic silicate rock resource base, converting that resource into a reserve through timely planning consent is a critical prerequisite for any scale-up. Incorporating this reality into our discussion underscores that, alongside infrastructure and logistical considerations, accelerating permit approvals for both existing-quarry expansions and new or reactivated sites will be essential to secure the reserves needed to meet future ERW and aggregate demand. Policy measures to streamline and prioritise these consents will therefore be pivotal to achieving the upscaling pathways we propose.

Moreover, as part of efforts for obtaining planning consents, public acceptability[21,22] and regulatory endorsement[42] are critical to any quarry expansion. Madankan & Renforth[20] quantified population proximity to UK basic silicate quarries as a proxy for social impact and showed that over 85% of current production and reserves lie in sites with below the UK's average social impact scores, implying that these considerations were largely addressed during initial planning consents. As highlighted by Bloodworth et al.[31], demographic pressures in the UK have substantially influenced public perception and acceptance of mining and quarrying activities. These concerns, along with the interplay between mining, other land uses and official designations such as historical or environmentally protected areas, shape the spatial planning system. These suggest that expanding currently active quarries may be more feasible than opening new sites, as the potential environmental and social constraints have already been addressed during the initial authorisation of these sites despite permission is still required for the expansion of existing quarries.

Expanding currently active quarries can be more straightforward due to existing infrastructure like access roads and processing facilities. These sites could scale up production with added machinery and workforce, focusing on capacity enhancement and logistical improvements, providing that there is sufficient working space within the site. In contrast, establishing new quarries requires substantial site investigation, (geological surveys, environmental assessments) and substantial infrastructure development, including constructing access roads, installing equipment and site preparation. Reactivating inactive quarries, while less complex than establishing new ones, still involves substantial technical and logistical efforts to bring them back into full operation. According to the Mineral Products Association report[41], crushed-rock planning applications take on average three years to determine, and the full development cycle, from initial exploration through to operation, can span 5–15 years. Moreover, the Aggregate Minerals Survey for England and Wales[28] shows that most existing crushed rock permissions are set to expire around 2040. These extended timelines underscore the necessity of submitting planning applications and renewals well in advance of projected demand.

Given these considerations, the feasibility of up-scaling rock supply for ERW may favour the expansion of existing quarries over the establishment of new ones. However, balancing the need for substantial capacity increases with the practical challenges of regulatory compliance, environmental sustainability and social acceptance remains crucial.

While this study outlines routes for upscaling of the upstream supply chain for ERW to achieve CDR targets, it reveals the necessity for further assessment of the trade-offs involved in each scenario. Balancing CDR potential, logistical efficiency, market dynamics and social and environmental impacts along with practical feasibility is essential to devise an effective strategy for ERW deployment. The findings emphasise the importance of strategic planning and robust stakeholder engagement to navigate the complexities of quarry expansion, ensuring

that the chosen scenario aligns with the overarching goals of environmental sustainability.

This case study employs the UK's basic silicate rock quarry production data from Madankan and Renforth[20]. While quarry capacities may have undergone modest changes since then, we anticipate that such updates would not substantially alter our core findings. While the UK provides a useful case study due to its specific geological and logistical context, the framework developed here can be adapted to other nations with suitable basic silicate rock resources and croplands. Countries like the United States, China, Brazil and India with globally substantial CDR potential through ERW[8] could leverage this model to explore their own ERW potential, adjusting for local variables such as the CDR potential of ERW on cropland, existing infrastructure, land use practices and regulatory environments. This adaptability underscores the broader applicability of our findings, offering a strategic blueprint for nations seeking to integrate ERW into their climate policies.

Effectiveness of the framework introduced in this study, is constrained by the availability and resolution of input data. For instance, it does not capture site-specific environmental or social impacts, such as local biodiversity sensitivities or community constraints, because such detailed datasets are often unavailable and require site-level assessments. The framework is nonetheless readily extensible by supplying the model with those data as additional constraints and directly incorporating site-specific impact criteria into the allocation algorithm.

While our spatio-temporal allocation model optimises for maximum $CO_2$ removal, in practice, economic considerations may also govern the allocations. The framework is extensible to accommodate such factors through some minor adaptations and using reliable cost input data. This can be done either by replacing the CDR-based ranking of quarry–cropland pairs with a cost-based ranking or by imposing cost-threshold constraints that enable the model to identify economically viable supply-chain pathways. Likewise, although we have employed an exogenous rock-supply target to estimate CDR potential, the full set of spatial and temporal allocation outputs from our model could be applied 'in reverse' to support removal objectives. By aligning the optimised allocations with a specified annual CDR trajectory, one could infer the required rock extraction schedules and allocation plans. Integrating this reverse-mapping approach, alongside economic assessments, offers a promising avenue for future research, helping to translate our initial CDR-focused framework into practical deployment planning.

## Methods
### Input data
Required data for this study were obtained from several sources. While some data on the aggregate production is publicly available in the UK, such as the Aggregate Minerals Survey for England and Wales[28], the lithology classification was too broad (limestone, sandstone, igneous and metamorphic) and regional aggregation lacks the mineralogical and spatial resolution required for our purpose. Instead, the data of the active and inactive basic silicate rock quarries in the UK, their geochemistry data, gridded map of the UK's croplands and transport distances between rock resources to cropland were directly adopted from our previous work[20]. For more details on data sources underlying our resource analysis, including historical production figures, as well as the full data collection methods, key assumptions and associated uncertainty bounds, please see our earlier study[20], its Supplementary information, and the references cited therein. It should be noted that the gridded map of UK croplands was filtered to omit grasslands due to the lack of precise CDR data that we later merged to form[18]. The resulting map includes arable cropland, which is expected to offer better accessibility and infrastructure readiness for rock spreading. The GB's 1:50,000 geological map[43] and 1:250,000 Geological map of the NI[44] was used to identify the potential location for new quarries. Target rock demand by 2070 was adopted from the rock supply scenarios, S1 (low rock supply), S2 (medium rock supply) and S3 (high rock supply) from ref. 18. Historical data of the

UK's domestic freight was obtained from the domestic freight transport by mode report[45] and the NI ports traffic report[27].

### Dedicated shipping routes
To minimise the $CO_2$ emissions associated with transporting basic silicate rock from NI to croplands in GB, we updated the transport distance matrix presented in our previous work[20] by introducing new sea shipping routes (Supplementary Information Fig. S4). This helps to meet the substantial demand of rock for croplands in England, which are often located far from quarries. To address this, we identified three strategically located importing ports in England (Liverpool, Avonmouth and Felixstowe) that could serve as distribution hubs for nearby croplands. Sea shipping was chosen as a low-carbon alternative for long-distance transport, as its $CO_2$ emissions and costs are substantially lower compared to road transport[16].

The updated transport matrix was developed using the same methodology described in our previous work[20] for calculating road transport distances. We focused on road transport because rail capacity for aggregate freight is currently limited, and over 87% of aggregate shipments already move by road[46]. Moreover, rail depot infrastructure for aggregates in Scotland and NI, regions from which most of the rock is sourced, is also constrained[47]. First, we calculated the road transport distances from each quarry in NI to the exporting port (Larne). Second, we calculated the road transport distances from each importing port in GB to all croplands. Third, the total transport distance for each quarry-cropland pair was determined by summing the corresponding road transport distances on both sides of the sea route. For each pair, three potential routes (via Liverpool, Avonmouth, or Felixstowe) were evaluated, and the route with the lowest total $CO_2$ emission from transport was selected. The $CO_2$ emissions for each quarry-cropland pair were calculated by adding the road transport emissions to the emissions from sea shipping (as detailed in Supplementary Information Table S3). This process ensured an optimised and low-emission logistics framework for basic silicate rock transport.

### Scenario development
In the low supply scenario (S1), the target demand is minimal, allowing it to be met solely through the expansion of currently active quarries without requiring the reactivation of inactive quarries or the opening of new sites. For the medium supply scenario (S2), two strategies were considered: S2a, which involves expanding the capacity of active quarries up to 2 Mt yr$^{-1}$ per quarry, and S2b, which limits quarry capacity to 1 Mt yr$^{-1}$ per quarry but necessitates the reactivation of inactive quarries to meet the demand. For the high supply scenario (S3) three strategies were explored: S3a, which relies exclusively on expanding active quarries with a substantial cap increase to up 5 Mt yr$^{-1}$ per quarry; S3b, which combines the expansion of active quarries with the reactivation of inactive ones, reducing the cap to 2 Mt yr$^{-1}$ per quarry; and S3c, which maintains a cap of less than 1 Mt yr$^{-1}$ per quarry, requiring the expansion of active quarries, reactivation of inactive quarries and the opening of new quarries to meet the high demand. These scenarios and their sub-strategies—S1, S2a, S2b, S3a, S3b and S3c—were designed to explore different pathways for scaling up rock supply.

### Finding potential new quarry locations
To determine the potential locations for new quarries, geological units with lithologies suitable for ERW were first extracted from the geological map using QGIS. This extracted map was then overlaid with both the map of the UK's protected areas from our previous study[20] and the UK's Land Cover map[48,49] to exclude urban and protected regions. The remaining suitable areas were converted into a 10 × 10 km grid map. The centroids of these grids were identified as potential new quarry sites, as illustrated in Fig. S3 in the Supplementary information. While exact reserve data is not available for potential new quarries, by assuming 1 km$^2$ of extractable outcrop with at least 50 m thickness within each 10 × 10 km (100 km$^2$) grid cell, then at a

typical silicate-rock density of 2.7 t m$^{-3}$, this corresponds to about 135 Mt of extractable rock for each suggested potential new quarry.

## Spatio-temporal allocation model
Upscaling the upstream rock supply chain for ERW necessitates a robust planning to effectively allocate the expanded capacity across both geographical and temporal dimensions. The spatio-temporal allocation model developed in this study is designed to optimise the upscaling of rock extraction, ensuring an efficient resource extraction plan as well as the effective allocation of resources to croplands over time for the deployment of ERW.

The first step in the methodology involved constructing a comprehensive dataset of all potential rock sources (active, inactive and new quarries) and croplands across the UK. Each quarry and cropland were then positioned at opposite ends of a matrix. The model involved calculating the net CDR potential (Eq. 1) between each pair of quarries and croplands.

$$NetCDR = \left(R_{CO_2} \times CDR_{Cropland}\right) - \left(CO_2\,e\,m + CO_2\,e\,g + CO_2\,e\,t + CO_2\,e\,s\right)$$
(1)

In the above equation, the $R_{CO_2}$ is the maximum $CO_2$ capture potential of a rock type based on its geochemistry (Renforth, 2012) and $CDR_{Cropland}$ is the rate at which $CO_2$ is sequestered in the cropland as a proportion of the $R_{CO_2}$. This factor accounts for the weathering efficiency of the rock under the specific soil and environmental conditions of the cropland. The $CDR_{Cropland}$ was determined using model outputs from Kantzas et al.[18] for croplands in GB. We calculated the maximum $CO_2$ capture potential of basic silicate rock used in their model ($R_{CO2}$) using the provided geochemical data. We then calculated a ratio by dividing the gross CDR of croplands in their model by this $R_{CO2}$, yielding a value that represents the proportion of the maximum CDR potential of basic silicate rock that can be effectively exploited within each cropland grid cell. This calculation accounts for various factors, including soil properties, environmental conditions, weathering efficiency and climate variability, as outlined in the CDR model by Kantzas et al.[18]. By incorporating these parameters, our model ensures that the final Net-CDR calculation reflects both the spatial variability in rock geochemistry ($R_{CO2}$) and the CDR rate across different croplands.

The $CO_2\,e\,m$, $CO_2\,e\,g$, $CO_2\,e\,t$ and $CO_2\,e\,s$ are the $CO_2$ emissions associated with rock extraction, grinding, transport and spreading onto the cropland, respectively. Table S1 in the Supplementary information summarises the energy consumption values used in our model for these processes. These secondary emissions were calculated by multiplying the energy consumption associated with each of these process (in kWh) by the projected life cycle emissions (in grCO$_2$ kWh$^{-1}$) in the UK from 2025 to 2070 derived from Kantzas et al.[18].

After calculating the Net CDR for each quarry-cropland pair, the results were compiled into a list. This list, containing all pairs along with their associated Net CDR values, was then sorted in descending order based on their CDR. The purpose of this sorting is to establish a priority for allocating the expanded rock capacity, ensuring that the quarries and croplands with the highest Net CDR are prioritised, thereby maximising the overall CDR efficiency.

Additional necessary data for the allocation model, such as annual rock demand for each cropland (calculated by multiplying the application rate of 40 t ha$^{-1}$ by cropland's area) and the annual supply capacity of each quarry (calculated based on the current base capacity and the defined annual growth rate) were added into the list. This finalised list serves as the primary input for the spatio-temporal allocation model. The other input was the target annual rock supply for each scenario.

With the input data in place, the model begins by allocating the expanded rock capacity to the prioritised quarry-cropland grid pairs. It starts with the higher-priority pairs, at top rows of the sorted list, fulfilling the cropland grid's rock demand using the available supply from the corresponding quarry. After each allocation, the model updates the supply capacity of the quarry and the demand of the cropland for that specific pair. If either the quarry's capacity or the cropland's demand is equal to zero, the subsequent rows involving that quarry or cropland with zero capacity or demand are omitted from the list for that iteration year. If there is any remaining capacity at the quarry or unmet demand at the cropland, these will be considered in the following rows. This process continues down the list, pair by pair, until the cumulative allocated rock supply for that year reaches the target. The transport distance constraint is enforced in this process so that any quarry-cropland pair whose transport distance exceeds a specified threshold is omitted from consideration. This threshold can be adjusted for different study regions; in the UK case study, we set it to 500 km. The model repeats this allocation process for each year, adjusting quarry capacities according to the defined annual growth rates and ensuring that the rock is distributed efficiently to maximise CDR across the croplands.

After the model is executed, key metrics are extracted to evaluate the performance of different scenarios. These metrics include the Net CDR, total rock extracted, transport capacity required, number of supplying quarries and CDR efficiency, all of which are analysed in both annual and cumulative formats for years 2025–2070.

Uncertainty analysis was undertaken to quantify the influence of lithological heterogeneity on the model outputs. We focused on the $R_{CO2}$ parameter and calculated its coefficient of variation (~11%) from the $R_{CO2}$ data[20] for all rock units in this study. Baseline $R_{CO2}$ values assigned to every quarry were therefore rescaled by ±11% to define 'high' and 'low' cases, while all other inputs were held constant. The model was executed for the baseline, high and low configurations and the resulting spread in annual net CDR, transport demand and CDR rate is presented as the shaded envelope in Fig. 1. This procedure provides a first-order, data-driven estimate of uncertainty attributable to rock geochemistry.

## Mapping the results
To assess key metrics such as annual rock extraction and total reserve requirements by UK counties, the outputs from the spatio-temporal model were integrated with the spatial layer of quarries and overlaid onto a UK counties map using QGIS v.3.22. We calculated extraction density, a metric that represents the cumulative reserve requirement relative to the area of each county, by dividing the total projected rock extracted from each county between 2025 and 2070 by the county's area. This metric provides an estimate of the potential environmental and social impacts of intensified rock extraction, highlighting areas where extraction activities are most concentrated and where impacts may be more pronounced. All results were visualised using QGIS v.3.22.

## Data availability
The UK cropland map was obtained from https://www.ceh.ac.uk/ukceh-land-cover-maps. The domestic freight transport data of the UK are available from the Department for Transport at https://www.gov.uk/government/statistical-data-sets/tsgb04-freight, and Northern Ireland ports data are available at https://www.nisra.gov.uk/publications. Local authority district's boundaries map of the UK is available at: https://geoportal.statistics.gov.uk/datasets/ons::local-authority-districts-may-2024-boundaries-uk-bfe-2/about. The full quarry dataset was obtained under licence and cannot be publicly shared.

## Code availability
The spatio-temporal allocation model used in this study is developed in Python v.3.11 and can be available upon reasonable request from the corresponding author.

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

## Acknowledgements

This work was supported by funding from UKRI for the Greenhouse gas removal with UK agriculture via enhanced rock weathering project (BB/V011359/1).

## Author contributions

M.M. contributed to study design, model development and implementation, data collection and analysis, and drafted the manuscript. E.K. contributed to data provision and manuscript review. R.E., S.V., L.K., P.S. and D.B. reviewed the manuscript and provided suggestions. P.R. guided the study and contributed to its conceptual development, manuscript preparation and review.

## Competing interests

D.J.B. has a minority equity stake in companies (Future Forest/Undo) and is an advisory board member of The Carbon Community, a UK carbon removal charity.
