## [Transparent Peer Review file · Communications Earth & Environment]

Larger rock extraction sites could improve the efficiency of enhanced rock weathering in the United Kingdom

Corresponding Author: Professor Phil Renforth

Version 0:

Decision Letter:

Dear Professor Renforth,

Your manuscript titled "A framework for national resource assessment for enhanced rock weathering, a UK national case study" has now been seen by 2 reviewers, and we include their comments at the end of this message. They find your work of interest, but some important points are raised. We are interested in the possibility of publishing your study in Communications Earth & Environment, but would like to consider your responses to these concerns and assess a revised manuscript before we make a final decision on publication.

We invite you to submit a revised manuscript that offers a compelling allocation model for a supply of rock for enhanced rock weathering that engages more comprehensively with relevant source data, transport-related constraints, and the associated environmental and social impacts. Please acknowledge your modeling assumptions and be transparent about the model's limitations—including regarding its scalability. We recommend expanding on the feasibility of this framework's implementation in the discussion section.

Please submit a revised manuscript, along with a point-by-point response that takes into account the points raised. Please highlight all changes in the manuscript text file.

Please submit your point-by-point responses as a separate file, distinct from your cover letter where you can add responses to the Editors' comments that you do not want to be made available to the reviewers. Word files are preferred. We recommend that any figures, tables or graphs that are included in the response to reviewers are also included in the main article or Supplementary Information.

Please use the following link to submit your revised manuscript, point-by-point response to the referees' comments (which should be in a separate document to any cover letter), a tracked-changes version of the manuscript (as a PDF file) and the completed checklist:

Link Redacted

We hope to receive your revised paper within six weeks; please let us know if you aren't able to submit it within this time so that we can discuss how best to proceed. If we don't hear from you, and the revision process takes significantly longer, we may close your file. In this event, we will still be happy to reconsider your paper at a later date, as long as nothing similar has been accepted for publication at Communications Earth & Environment or published elsewhere in the meantime.

Please do not hesitate to contact us if you have any questions or would like to discuss these revisions further. We look forward to seeing the revised manuscript and thank you for the opportunity to review your work.

Best regards,

Yann Benetreau, PhD
Consulting Editor, Communications Earth & Environment
Deputy Editor, Communications Sustainability
Nature Portfolio
ORCID: 0000-0002-1897-0887
New York Office

EDITORIAL POLICIES AND FORMATTING

Editorial Policy: [Policy requirements](https://www.nature.com/documents/nr-editorial-policy-checklist.pdf) (Download the link to your computer as a PDF.)

- Behavioural and social science
- Ecological, evolutionary & environmental sciences
- Life sciences

<https://www.nature.com/documents/nr-reporting-summary.zip>

Furthermore, please align your manuscript with our format requirements, which are summarized on the following checklist: [Communications Earth & Environment formatting checklist](https://www.nature.com/documents/commsj-phys-style-formatting-checklist-article.pdf)

and also in our style and formatting guide [Communications Earth & Environment formatting guide](https://www.nature.com/documents/commsj-phys-style-formatting-guide-accept.pdf).

***** DATA:** Communications Earth & Environment endorses the principles of the Enabling FAIR data project (<http://www.copdess.org/enabling-fair-data-project/>). We ask authors to make the data that support their conclusions available in permanent, publically accessible data repositories. (Please contact the editor if you are unable to make your data available).

All Communications Earth & Environment manuscripts must include a section titled "Data Availability" at the end of the Methods section or main text (if no Methods). More information on this policy, is available at <http://www.nature.com/authors/policies/data/data-availability-statements-data-citations.pdf>.

If a community resource is unavailable, data can be submitted to generalist repositories such as [figshare](https://figshare.com/) or [Dryad Digital Repository](http://datadryad.org/). Please provide a unique identifier for the data (for example a DOI or a permanent URL) in the data availability statement, if possible. If the repository does not provide identifiers, we encourage authors to supply the search terms that will return the data. For data that have been obtained from publically available sources, please provide a URL and the specific data product name in the data availability statement. Data with a DOI should be further cited in the methods reference section.

REVIEWER COMMENTS:

Reviewer #1 (Remarks to the Author):

The paper presents an interesting analysis of how a new mechanism for CCS may be implemented. However, although I appreciate the paper has a limited scope and cannot cover all aspects there are significant gaps in the source data used for modelling and aspects of the mineral supply and planning process in the UK that currently limit many of the papers conclusions to those of an interesting thought experiment and lack a realistic commentary on how the enhanced rock weathering in the UK may (or may not) be progressed.

I also do not think the paper adequately addresses the associated environmental impacts of mineral extraction on the scale they are suggesting may cause. Quarrying is an often controversial land use and will often lead to biodiversity loss, as well as the act of quarrying, crushing rocks and transporting material having a significant footprint in terms of emissions itself. Although I appreciate this is not the focus of the paper it is something that does need to be addressed due to the sensitivities around suggesting such significant volumes of extraction. For example, please can the authors explicitly state they have included the emissions from the extraction and transport of crushed rock in their calculations? The reference of Reershemius and Surhoff 2024, which is used with regard emissions from ERW does not contain this information. A short review of the literature around LCA and ERW and how its effectiveness for CO₂ reduction is dependent on supply chains, energy sources, and transport distances may be helpful.

Significant sources of information that are directly relevant to the study appear to be missing. Such as the Aggregate Mineral Survey, which gives detailed breakdowns of aggregate resources production, reserves, by region, including interregional flows. <https://www.gov.uk/government/publications/aggregate-minerals-survey-for-england-and-wales-2019>

Data for Northern Ireland mineral production, much of this study is around increasing production from NI, this series details the basalt production and should be included: <https://www.economy-ni.gov.uk/publications/annual-minerals-statements>

Data from the Mineral products association for transport and freight

<https://www.mineralproducts.org/Sustainability/Reporting.aspx> and more importantly the data for the replenishment of resources, https://www.mineralproducts.org/MPA/media/root/News/2024/AMPS_Report_2023.pdf

This observed data seems to contradict some of the model results. i.e. the UK is seeing a reduction in available reserves that is set to continue and constrain supply.

Data for the location of UK mineral extraction from the BGS BritPits database (this has been used but seeming incorrectly).

The BGS Directory of Mines and Quarries may also be of use <https://www.bgs.ac.uk/mineralsuk/minerals/mine-and-quarry/directory-of-mines-and-quarries/>

Data for future policy related to minerals demand. i.e house building and infrastructure targets. These suggest the demand for crushed rock will outstrip supply. This seems to contradict statements in the manuscript that there will be abundant supply for ERW. Another important planning consideration is the end date of many crushed rock site permissions, this data can be found in the Aggregate Mineral Survey.

Other comments:

I am confused how will large extraction sites increase transport efficiencies, surely the fewer quarries the larger the distance between source and supply? This can be seen in practice when looking at how the average road aggregate transport distances significantly increased post Covid as many sites were mothballed and the remaining sites picked up supply (this transport distance data is contained within the Mineral Product Association sustainability reports). Please can this be detailed further.

Also regarding transport distances the data contained within the MPA sustainability reports state the average distance for transport of crushed rock as 28 miles by road (this includes sand and gravel, so will be a little high for crushed rock) but how does these small distances factor into the models proposed by tis paper for many hundreds of miles as well as shipping?

Similarly there is discussion (line 184) of imports from Northern Ireland. Currently only high specification aggregates for use in skid resistant road surface are imported, due to the high cost associated and limited resources in GB. Proposing a significant increase from NI for materials with much lower prices per tonne seem very unrealistic and require further explanation (see point below regarding economic model). Specifically, I note the study suggests imports from NI to England of 66.7mt, currently it is 'a small fraction of 4.8mt (total imports) source: Aggregate Minerals Survey. Is this realistic? The construction of the transport and port infrastructure alone would seem to preclude it and need to be adequately acknowledged.

The economic model for this extraction is not mentioned? As can be seen in the reduction of soil treatments, such as liming, a very similar process to the ERW being discussed (<https://www.gov.uk/government/collections/fertiliser-usage>), there is little incentive for farmers to purchase such material. Will this be a taxpayer funded scheme? What is the suggested business model?

Line 39, whilst the UK may have abundant basalt in Northern Ireland and some parts of Scotland it is only an abundant rock type in locations that are distant to the main areas of managed croplands in the UK. There is almost no basalt outside these areas, i.e. almost all of England and Wales. This is a significant limitation on its utilization. Perhaps the authors do not necessarily mean basalt but more generally all Mg and Ca rich silicate rocks? (again limited distribution for most of England, for example see figure 7 <https://nora.nerc.ac.uk/id/eprint/524079/>). This geological confusion occurs several times in the manuscript. For example line 45 discussing the UK basalt production, this should read basic igneous rock. This is important as these different rock types will have significantly different reactivities. I also note that the study that has been used for the basic silicate rock production and reserve figures, Madankan and Renforth base there data on the British geological Survey BritPits database stating that "Given that the lithology of the quarry's product is not identified in BritPits,". The Lithology is included within this database, and the omission of this reduces confidence in the estimates that have been made for crushed rock production and reserves data used. This data can also be found in the Aggregate Minerals Survey for England and Wales.

There are many very significant assumptions made in the study regarding almost all aspects of the crushed rock supply chain. I note the BGS UK Minerals yearbook does not attempt to break down crushed rock by rock type as due to the considerable errors involved. Some acknowledgement of these assumptions and affects on the model would be welcome.

Reviewer #2 (Remarks to the Author):

Review for the manuscript "A framework for national resource assessment for enhanced rock weathering, a UK national case study".

This paper presents a comprehensive workflow to match rock resources with transport and croplands for spreading crushed silicate rocks for enhanced rock weathering, applied to the UK. The presented spatio-temporal allocation model is a fairly easy-to-understand allocation model that works effectively for this top-down approach and at this early level of development (but also see comment 2). The model considers the main important data, needed to make such ERW assessment. The data used also seems complete. The paper is well-structured, well-written and is mostly easy to follow.

There are, however, a few issues that need attention.

1. I suggest changing the title. It promises a resource assessment for ERW. A resource, however, is a quantity that could be available for use, while here a target for rock use is imposed and the amount of ERW is calculated. The paper rather presents a "source-sink" matching.
2. In general, I find it unclear how uncertainties are introduced and used, although from Figure 1 it is clear that there is some influence on the results.
3. This is very much a top-down approach, allocating resources as a governing body (which is probably ok an early assessment). But in reality economic decisions will determine which match will be made, which will need an economic assessment/match too. I suggest adding this to the discussion. Similarly, now the target is an amount of rock supply, used to calculate ERW potential, because these numbers emerged from another study. For a first assessment this could be an ok approximation, but for emission reduction goals it should be reversed.
4. While the positive impact to the soil (CO₂ uptake) is the main metric that is used and shown, potential negative effects are not mentioned (e.g. heavy metals). Considering that the framework is in place for the positive geochemical impact, I imagine it should not be very difficult to also calculate other (potentially negative) impacts? Although it would be nice to see such environmental analysis here, it would be sufficient if this is included in the discussion.
5. Another suggestion which could be very interesting for investors and policy makers, is to overlay the extraction map with population density, which provides a first social impact assessment.
6. I suggest changing the scenario names. Although they are concise, there is no relation to the actual scenario content, hence it is more difficult for readers to follow.
7. When introducing climate change in lines 24-29, the most recent citation is 2022. As research is moving fast, I suggest adding more recent literature. Moreover, the 1.5°C target is already passed, so mentioning the timeline of "climate" is essential.
8. Line 48: It is mentioned that such a rock extraction increase is not unprecedented, but there is no mention of when or where this happened (or a reference).
9. Lines 58-61 say that there is a need for including environmental and social considerations, and that this study aims to address this need. But this paper does not include an environmental or social study? The problem statement/goals should therefore be rephrased.
10. In my opinion it is not very clear from the start what the scenarios and their constraints are. This is somewhat scattered throughout the paper I suggest adding a table with the major parameters.
11. The result differences between the scenarios, presented in Table 1, are caused by two parameters: the production amount+quarry activation, and the cap per quarry. I suggest to present and discuss the results much more in function of these parameters. For example, it is unclear why the CDR rate is decreasing with increasing scenario freedoms (S1->S3). Is it caused by the lower cap per quarry?
12. For figure 2, I suggest to join the legend components together, and explicitly mention what the area is for the rock spread amount (10*10km I assume).
13. I suggest adding scenario S3.a to Figure 3 anyway, which will show the evolution over the scenarios even more clearly.
14. In Figure 4, the counties are used as cells for density calculation. However, using county-sized areas provides a very disturbed picture. For example: if there is one large quarry, for a large county it would show up lightly colored, while for a small county it would be darkly colored. I suggest to remake these maps with a uniform grid, e.g. 10*10km (see also lines 466-471).
15. Figure 5: please indicate in the caption that these results are from Scenario Sc3.c.
16. Figure 5b: Considering the freight growth since the 1960s, I'm not sure if assuming a stable amount of transport for the coming 50 years is realistic, just because it has been stable for last couple of years. Moreover, in lines 288-289 this leads to a false statement: it is a very rough assumption that freight amounts will stay stable. Drawing the conclusion that the additional required capacity is lower than the historical maximum is wrong.
17. Line 295: Regarding the reference for the cost of port capacity increase: is there no example from the UK (or Europe) that is more representative?
18. Line 312: The discussion on the environmental and social impact ends quite abruptly, without having a proper discussion on environmental benefits vs downsides, social impact (equitable distribution...).
19. Line 330-331: Also consider permitting in the lead time.
20. Line 338 states that this study outlines routes for sustainable upscaling, but sustainability is not evaluated or taken into account in the model or matching. Especially marketing and popular media are very uncaringful in using "sustainability", I urge to be much more careful in science.
21. As far as I can find, when identifying potential new quarry areas only the surface area (from geological maps) are is

considered. However, thickness of the suitable formations is as important, has there been no reserve calculation for the UK? 22. Line 427-428: Does the CDR potential calculation also account for the relative amount of suitable cropland per grid cell? Mention this explicitly if so.

23. The publication by Euripides et al. (2022), cited in lines 429 and 437, is not in the reference list, while it is a very important paper for reaching the current results.

24. In Madankan & Renforth (2023), maps are published on the relative appropriateness of UK croplands for ERW (Figure 5 of that paper). It is very apparent that the most suitable areas from that paper are opposite to those of the current paper (e.g. Figure 2, partly also Figure S4). Since this paper shares much data and methods with the Madankan & Renforth paper, why is this difference so apparent? I would at least mention this somehow in the methods or discussion.

Considering these comments, I recommend publishing this manuscript, though with a major review.

Communications Earth & Environment is committed to improving transparency in authorship. As part of our efforts in this direction, we are now requesting that all authors identified as 'corresponding author' create and link their Open Researcher and Contributor Identifier (ORCID) with their account on the Manuscript Tracking System prior to acceptance. ORCID helps the scientific community achieve unambiguous attribution of all scholarly contributions. You can create and link your ORCID from the home page of the Manuscript Tracking System by clicking on 'Modify my Springer Nature account' and following the instructions in the link below. Please also inform all co-authors that they can add their ORCIDs to their accounts and that they must do so prior to acceptance.

Version 1:

Decision Letter:

Dear Professor Renforth,

Your manuscript titled "A spatio-temporal supply-chain framework for Enhanced Rock Weathering deployment at scale: a UK case study" has now been seen by our reviewers, whose comments appear below. In light of their advice we are delighted to say that we are happy, in principle, to publish a suitably revised version in Communications Earth & Environment.

We therefore invite you to revise your paper one last time to address the remaining concerns of our reviewers. At the same time we ask that you edit your manuscript to comply with our format requirements and to maximise the accessibility and therefore the impact of your work.

EDITORIAL REQUESTS:

****Please take care to match our formatting and policy requirements. We will check revised manuscript and return manuscripts that do not comply. Such requests will lead to delays. ****

SUBMISSION INFORMATION:

OPEN ACCESS:

Communications Earth & Environment is a fully open access journal. Articles are made freely accessible on publication. For further information about article processing charges, open access funding, and advice and support from Nature Research, please visit <https://www.nature.com/commsenv/open-access>

Link Redacted

Best regards,

Yann Benetreau, PhD
Consulting Editor, Communications Earth & Environment
Deputy Editor, Communications Sustainability
Nature Portfolio
ORCID: 0000-0002-1897-0887
New York Office

REVIEWERS' COMMENTS:

Reviewer #1 (Remarks to the Author):

Thank you for your comprehensive response to the comments made. I am happy to recommend to the editor the manuscript is accepted.

Reviewer #2 (Remarks to the Author):

I would like to thank the authors for considering my comments and suggestions, and adjusting the manuscript accordingly. I have three minor comments left that relate to form rather than content:

In figure 1, only the first sub-figure received an (a) label; b, c and d are missing.

The sentence in line 378 has, I believe, a grammar issue.

The reference in line 706 is written incorrectly. This is a report for the UK government, and although there are no authors mentioned, I think the reference should be: "Element Energy & UK Centre for Ecology & Hydrology, 2021. Greenhouse gas removal methods and their potential UK deployment. Report for the UK Department for Business, Energy and Industrial Strategy, 105 p.

Other than that I recommend this manuscript for publication.

REVIEWER COMMENTS:

Reviewer #1

The paper presents an interesting analysis of how a new mechanism for CCS may be implemented. However, although I appreciate the paper has a limited scope and cannot cover all aspects there are significant gaps in the source data used for modelling and aspects of the mineral supply and planning process in the UK that currently limit many of the papers conclusions to those of an interesting thought experiment and lack a realistic commentary on how the enhanced rock weathering in the UK may (or may not) be progressed.

We thank the reviewer for recognising the paper's focused scope. Our primary aim is to develop and demonstrate a nation-adaptable framework for assessing upstream rock supply and its scalability for ERW through a spatio-temporal allocation model.

With the UK case study, our primary aim is to demonstrate how different up-scaling pathways for ERW might look like from an upstream rock-supply perspective, using a detailed spatio-temporal allocation model. We have mapped the spatial and temporal expansion of quarry capacity required to meet various CO₂-removal targets as accurately as possible, and flagged key real-world constraints, permit lead times, transport logistics, infrastructure requirement, environmental and social impacts, to alert readers to potential limitations. A definitive judgement on feasibility depends on policy-making and regulatory actions beyond this study's remit. Instead, our scenarios offer policymakers the insights on the trade-offs they need to know when deciding at what scale, and under what conditions, ERW might be implemented. We address the reviewer's more specific concerns below.

I also do not think the paper adequately addresses the associated environmental impacts of mineral extraction on the scale they are suggesting may cause. Quarrying is an often controversial land use and will often lead to biodiversity loss, as well as the act of quarrying, crushing rocks and transporting material having a significant footprint in terms of emissions itself. Although I appreciate this is not the focus of the paper it is something that does need to be addressed due to the sensitivities around suggesting such significant volumes of extraction.

Thank you for raising the need to more explicitly acknowledge quarrying's environmental footprint at scale. In response, we have added two complementary passages, one in the main Discussion (highlighted in yellow under section: Discussion / Environmental and Social impact) and one in the Supporting Information (Highlighted under title of Projected Land-Use Change from Quarry Expansion and Fig. S5). We did new analysis quantifying the projected land use changes due to expansion of quarries under our most ambitious expansion scenario (S3.a). In addition, we reviewed several sources including papers and reports on potential impact of quarrying on biodiversity loss as well as opportunities for net-gain through robust environmental impact assessments and detailed biodiversity action plans.

For example, please can the authors explicitly state they have included the emissions from the extraction and transport of crushed rock in their calculations?

The CO₂ emissions associated with rock extraction, grinding, transport and spreading onto the cropland are all included in the calculation (please see the Eq.1 in the Methods section

and its following paragraphs). Also, Table S.1 in the supporting information summarises the energy consumption values used in our model for these processes. These secondary emissions were calculated by multiplying the energy consumption associated with each of these process (in kWh) by the projected life cycle emissions (in gCO₂ / kWh) in the UK from 2025 – 2070 derived from (Kantzas et. al., 2022).

The reference of Reershemius and Surhoff 2024, which is used with regard emissions from ERW does not contain this information.

Reference has been removed.

A short review of the literature around LCA and ERW and how its effectiveness for CO₂ reduction is dependent on supply chains, energy sources, and transport distances may be helpful.

We have added a concise literature review to the Introduction (highlighted in yellow) that summarises requested points from these studies:

- *Eufrazio et al. (2022). Environmental and health impacts of atmospheric CO₂ removal by enhanced rock weathering depend on nations' energy mix. Communications Earth & Environment.*

- *Foteinis et al. (2022). Life cycle assessment of ocean liming for carbon dioxide removal from the atmosphere. Journal of Cleaner Production.*

- *Beerling et al. (2024). Enhanced weathering in the US Corn Belt delivers carbon removal with agronomic benefits. PNAS.*

Lefebvre et al. (2019). Assessing the potential of soil carbonation and enhanced weathering through Life Cycle Assessment: A case study for Sao Paulo State, Brazil. Journal of Cleaner Production.

- *Kantzas, et al. (2022). Substantial carbon drawdown potential from enhanced rock weathering in the United Kingdom. Nature Geoscience.*

Significant sources of information that are directly relevant to the study appear to be missing. Such as the Aggregate Mineral Survey, which gives detailed breakdowns of aggregate resources production, reserves, by region, including interregional flows.

<https://www.gov.uk/government/publications/aggregate-minerals-survey-for-england-and-wales-2019>

Thank you for pointing us to the Aggregate Mineral Survey (GOV.UK, 2019), which indeed provides valuable region-level data on aggregate production and reserves. However, its three broad lithology classes (“limestone”, “sandstone” and “igneous & metamorphic”) and its England and Wales-only, regional aggregation do not meet the lithologic or spatial resolution needed to identify quarries optimally suited for enhanced rock weathering (ERW).

To overcome these limitations, we (Madankan & Renforth 2023) previously compiled our own quarry-by-quarry inventory for the entire UK. By cross-referencing each active site against the BGS 1:50 000 geological map, we extracted exact lithologies thought to be effective for ERW, and then directly assembled site-level production and reserves data from operator reports and MPA records. This bespoke dataset therefore offers both the detailed mineralogy and the full UK coverage that our ERW model requires.

Accordingly, we have added a brief note to the Methods explaining why the Aggregate Mineral Survey was too coarse for our purposes and directing readers to our quarry-level inventory. We also used the total aggregate consumption figure for England and Wales to compare it to the total rock supply needed for ERW by 2070 in discussion under the Practical feasibility of the rock supply up-scaling Section.

Data for Northern Ireland mineral production, much of this study is around increasing production from NI, this series details the basalt production and should be included:

<https://www.economy-ni.gov.uk/publications/annual-minerals-statements>

Production statistics for basalt and igneous crushed rock in Northern Ireland (from mentioned reports for years 2000–2021), together with the UK-wide crushed-rock series (1980–2020; UK Mineral Yearbook), were compiled and analysed in our earlier study (Madankan & Renforth 2023) and all references are available there. The present manuscript adopts that dataset to support the deployment scenarios produced by our spatio-temporal allocation model; we therefore do not reprise the descriptive resource analysis here. A note has been added to the Methods section directing readers to the previous paper and its supporting information for complete details of the underlying resource data. We also used the most recent basalt and total crushed rock production figure of Northern Ireland from this report to compare it to the 60 Mt/yr of supply needed for ERW by 2070 from Northern Ireland. This is highlighted in discussion section under Practical feasibility of the rock supply up-scaling.

Data from the Mineral products association for transport and freight

<https://www.mineralproducts.org/Sustainability/Reporting.aspx>

We've drawn on key figures from the MPA's Sustainability Reporting (Mineral Products Association, 2022) and their Rail Freight report (MPA, 2019), notably the average road transport distance of 28 miles and the fact that 87% of aggregates move by road in the UK. We've now clarified in the Methods and Discussion how these values link/compare to our transport-mode assumptions and average transport distances.

and more importantly the data for the replenishment of resources,

https://www.mineralproducts.org/MPA/media/root/News/2024/AMPS_Report_2023.pdf

This observed data seems to contradict some of the model results. i.e. the UK is seeing a reduction in available reserves that is set to continue and constrain supply.

We thank the reviewer for drawing our attention to the AMPS 2023 report on permitted reserves, which shows that planning approvals for crushed rock are falling behind demand and widening a “permit gap.” However, this finding does not contradict our model, which explores upscaling pathways based on the UK's abundant basalt “resources” rather than “reserves”. For clarity, “resources” refers to all identified basalt formations in the UK, while “reserves” denotes the subset of those formations with active extraction permits.

In our revised Discussion, we integrate the AMPS data to emphasise that, to realise the proposed upscaling pathways, planning applications must be submitted and approved in good time so that these plentiful resources can be converted into sufficient reserves to meet future demand. Moreover, the substantial carbon-removal benefits of this national-scale

ERW initiative should help streamline and prioritise permit approvals for such critical climate action.

Data for the location of UK mineral extraction from the BGS BritPits database (this has been used but seeming incorrectly). The BGS Directory of Mines and Quarries may also be of use <https://www.bgs.ac.uk/mineralsuk/minerals/mine-and-quarry/directory-of-mines-and-quarries/>

We thank the reviewer for suggesting the BGS Directory of Mines & Quarries (DMQ). However, the DMQ is derived from the BritPits database (explicitly stated in the link above) and offers no additional site records or spatial detail beyond the underlying BritPits GIS dataset.

In our previous work (Madankan & Renforth, 2023), we used BritPits as our primary source, rigorously cross-referencing each quarry location against the 1:50 k geological map and visually inspecting every site to confirm that only those producing lithologies suitable for ERW were included (as fully described in Madankan & Renforth, 2023).

We believe this approach ensures both completeness and accuracy. We would appreciate further clarification if the reviewer has identified any specific instances where our use of the BritPits data appears incorrect, so that we can address them directly.

Data for future policy related to minerals demand. i.e house building and infrastructure targets. These suggest the demand for crushed rock will outstrip supply. This seems to contradict statements in the manuscript that there will be abundant supply for ERW.

To clarify, our study does not assume that current supply is already abundant; rather, it maps out how to upscale supply beyond today's capacity. Specifically, we show that delivering sustained ERW at $\sim 30 \text{ Mt CO}_2 \text{ yr}^{-1}$ would require roughly a tenfold increase in present extraction rates by mid-century, underpinned by expanded permitted reserves. In our modelling framework, ERW feedstock supply is treated as an additional extraction on top of current aggregate demand, so the two demands are considered independently.

Underlying this is the fact that the UK's basalt resource base remains large, but converting it into working reserves depends on timely planning consents. We have therefore added text in the Discussion to emphasise this point (see our revisions addressing the earlier comment on reserve replenishment).

In direct response to this comment, we also clarify that if construction-aggregate demand were to rise in future, driven by housing or infrastructure targets, this could compete with ERW feedstock. However, the contribution of recycled and secondary materials in the UK aggregates market (which supplied about 28 percent, 69.6 Mt in 2021; MPA 2021) is increasing under circular-economy policies. This growth can help relieve pressure on primary extraction and thus enable both construction aggregate and ERW demands to be met concurrently.

Another important planning consideration is the end date of many crushed rock site permissions, this data can be found in the Aggregate Mineral Survey.

We have added a brief note in the Discussion highlighting that most crushed-rock permissions expire around 2040 (based on Aggregate Minerals Survey for England and

Wales (Mankelov et al., 2021) and that planning applications (including renewals) take several years (from 3-15 years) to determine and implement (based on AMPS 2023). This underscores the need to lodge consents well ahead of future ERW and aggregate demand.

Other comments:

I am confused how will large extraction sites increase transport efficiencies, surely the fewer quarries the larger the distance between source and supply? This can be seen in practice when looking at how the average road aggregate transport distances significantly increased post Covid as many sites were mothballed and the remaining sites picked up supply (this transport distance data is contained within the Mineral Product Association sustainability reports). Please can this be detailed further.

We understand the intuition that “fewer quarries” might suggest longer transport distances, but in our ERW context both the rock sources and the application sites are tightly constrained spatially. UK basalt resources lie mostly in Northern Ireland and the Central Belt of Scotland, while the croplands best suited for enhanced weathering are clustered in south-east England. Adding more small quarries to the supply network, many of which would be in northern Scotland, as shown in our Supporting Information, cannot necessarily bring rock closer to those fields.

The other points worth clarification (also added to the discussion section) is: The transport requirement is presented as t.km per tonne of CO₂ removed, so in scenario with fewer larger quarries, allowing quarries with more favourable conditions to contribute more to the rock supply, enhances CDR efficiency. The higher CDR efficiency offsets for the extra distance, reducing the overall transport footprint per unit CO₂ removed.

Also regarding transport distances, the data contained within the MPA sustainability reports state the average distance for transport of crushed rock as 28 miles by road (this includes sand and gravel, so will be a little high for crushed rock) but how does these small distances factor into the models proposed by tis paper for many hundreds of miles as well as shipping?

Similarly there is discussion (line 184) of imports from Northern Ireland. Currently only high specification aggregates for use in skid resistant road surface are imported, due to the high cost associated and limited resources in GB. Proposing a significant increase from NI for materials with much lower prices per tonne seem very unrealistic and require further explanation (see point below regarding economic model).

We have added further clarification to address this concern in the Discussion section. In the ERW context, the geographic distribution of supply and demand points (basalt resources and target croplands) differs markedly from conventional aggregate supply chains, where supply points can be chosen more flexibly and closer to demand points due to fewer rock-type constraints. In addition, basalt for ERW carries a much higher value per tonne (compared to conventional aggregates and can be compared more to high-spec aggregates as mentioned by the reviewer) thanks to its CO₂ removal potential and associated carbon credit revenue. Transporting rock hundreds of miles (or by ship) can make economic sense if the net carbon-credit income outweighs the additional transport costs. As Kantzas et al. (2022) demonstrate, transport remains a minor component of total ERW cost in their UK study.

Specifically, I note the study suggests imports from NI to England of 66.7mt, currently it is ‘a small fraction of 4.8mt (total imports) source: Aggregate Minerals Survey. Is this realistic?

The construction of the transport and port infrastructure alone would seem to preclude it and need to be adequately acknowledged.

We believe this concern has been already discussed in the discussion:

“Although the additional transport capacity required for ERW by 2070 will be lower than the historical maximum, country level breakdown reveals that Northern Ireland, as a primary supplier of rock for ERW by 2070, will need to export over 60 Mt of rock annually to England, Scotland and Wales. Presently 11.5 Mt of goods are exported through Northern Ireland’s ports, up from 5.5 Mt in 1998 (Northern Ireland Statistics & Research Agency, 2023). Given this context, a substantial increase to over 60 Mt annually necessitates future investment in shipping routes and port infrastructure to handle the increased volume efficiently. For example, a 6 Mt capacity enhancement at the Port of Melbourne cost approximately \$ 1.5 billion (Deloitte Access Economics. (2024). Applying benchmarks from bulk material ports, the CAPEX for a 60 Mt yr⁻¹ capacity port in Northern Ireland can be in the scale of tens billion dollars. Over a 50-year service life, the breakdown of this CAPEX per tonne of rock transported through this port can be approximately \$20 t⁻¹. “

We have also incorporated England and Wales’ aggregate imports (4.8 Mt, per the Aggregate Minerals Survey) into this discussion.

Moreover, the advantage of maritime shipping (low operational costs and CO₂ footprint compared to road haulage; Renforth, 2012), is discussed at the end of this section.

The economic model for this extraction is not mentioned? As can be seen in the reduction of soil treatments, such as liming, a very similar process to the ERW being discussed (<https://www.gov.uk/government/collections/fertiliser-usage>), there is little incentive for farmers to purchase such material. Will this be a taxpayer funded scheme? What is the suggested business model?

Note that designing an economic model for basalt extraction, including farmer uptake incentives, potential carbon-credit revenues, net-zero requirements for businesses and any taxpayer-funded subsidy scheme, falls outside the scope of this study. Currently, the voluntary carbon removal market and early stage investment has incentivised the addition of >100,000 tonnes of rock to cropland, including working with 1,000’s of smaller holder farmers in India (<https://www.mati.earth>) to larger plantations in Brazil (<https://inplanet.earth>).

Line 39, whilst the UK may have abundant basalt in Northern Ireland and some parts of Scotland it is only an abundant rock type in locations that are distant to the main areas of managed croplands in the UK. There is almost no basalt outside these areas, i.e. almost all of England and Wales. This is a significant limitation on its utilization. Perhaps the authors do not necessarily mean basalt but more generally all Mg and Ca rich silicate rocks? (again limited distribution for most of England, for example see figure 7 <https://nora.nerc.ac.uk/id/eprint/524079/>). This geological confusion occurs several times in the manuscript. For example line 45 discussing the UK basalt production, this should read basic igneous rock. This is important as these different rock types will have significantly different reactivities.

To avoid terminological confusion, we have replaced “basalt” with “basic silicate rocks.”

While these rocks occur mainly in Scotland and Northern Ireland, suitable resources do also exist in Wales and England (Fig. 2). Our study found an average transport distance to croplands of approximately 200 km for up-scaled ERW in the UK (Table 1). We were therefore surprised by the suggestion that this represents “a significant limitation on its

utilization,” since our model, which uses a high-resolution logistical analysis, still indicates a UK CDR potential of 5–24 Mt yr⁻¹ through ERW, findings that align with previous studies (Renforth, 2012; Beerling et al., 2020; Kantzas et al., 2022).

I also note that the study that has been used for the basic silicate rock production and reserve figures, Madankan and Renforth base their data on the British geological Survey BritPits database stating that “Given that the lithology of the quarry’s product is not identified in BritPits,”. The Lithology is included within this database, and the omission of this reduces confidence in the estimates that have been made for crushed rock production and reserves data used. This data can also be found in the Aggregate Minerals Survey for England and Wales.

BritPits’ rock type categorisation is limited to three broad categories (“limestone,” “sandstone,” and “igneous & metamorphic”). Igneous rocks include basic rocks, but also those that are unsuitable for enhanced weathering (e.g., granite). As such it is not possible to directly use BritPits.

In full, our description in Madankan & Renforth (2023) reads:

“Given that the lithology of the quarry’s product is not identified in BritPits, the dataset was cross referenced with the 1:50,000 geological map of the Great Britain (BGS, 2016) and 1:250,000 Geological map of the Northern Ireland (BGS, 2021). For this purpose, first, the map of geological units with potentially desired lithology for ERW were extracted from the geological map in QGIS. The details of selected geological units is provided in supporting information Fig. S1 and table. S1. Then, using the ‘join attributes by location’ feature of QGIS 3.22, each entry (quarry) of the filtered BritPits dataset obtained a corresponding lithology data from the lithology unit that spatially overlapped with it. The quarry entries which did not spatially overlap with the desired lithological units were omitted from the dataset.”

There are many very significant assumptions made in the study regarding almost all aspects of the crushed rock supply chain. I note the BGS UK Minerals yearbook does not attempt to break down crushed rock by rock type as due to the considerable errors involved. Some acknowledgement of these assumptions and effects on the model would be welcome.

As detailed earlier, in our previous work (Madankan & Renforth, 2023), our inventory of resources for ERW was built from BritPits quarry records, each location rigorously cross-referenced against the 1:50 k geological map and visually inspected to ensure only ERW-suitable lithologies were included; that paper also fully documents our data-collection protocols, key assumptions and uncertainty bounds. To avoid unnecessary repetition, we now explicitly point readers to that work in the Methods (and again in the Discussion), noting that all details of quarry selection, parameter assumptions, and their potential effects on model outputs are available there.

We also added this note in the discussion: It should be noted that this case study employs UK basalt quarry production data from Madankan & Renforth (2023). While quarry capacities may have undergone modest changes since then, we anticipate that such updates would not substantially alter our core findings.

Reviewer #2

Review for the manuscript “A framework for national resource assessment for enhanced rock weathering, a UK national case study”.

This paper presents a comprehensive workflow to match rock resources with transport and croplands for spreading crushed silicate rocks for enhanced rock weathering, applied to the UK. The presented spatio-temporal allocation model is a fairly easy-to-understand allocation model that works effectively for this top-down approach and at this early level of development (but also see comment 2). The model considers the main important data, needed to make such ERW assessment. The data used also seems complete. The paper is well-structured, well-written and is mostly easy to follow.

There are, however, a few issues that need attention.

1. I suggest changing the title. It promises a resource assessment for ERW. A resource, however, is a quantity that could be available for use, while here a target for rock use is imposed and the amount of ERW is calculated. The paper rather presents a “source-sink” matching.

While our framework indeed matches sources and sinks, our main contribution lies in the design of spatio-temporal deployment and up-scaling pathways for upstream rock supply. For this reason, we have adopted “supply chain” and “spatio-temporal” in the title. Given that this framework is applied directly to scalable ERW deployment, we have revised the title to:

“A spatio-temporal supply-chain framework for Enhanced Rock Weathering deployment at scale: a UK case study”

2. In general, I find it unclear how uncertainties are introduced and used, although from Figure 1 it is clear that there is some influence on the results.

We now quantify uncertainty by varying the key parameter RCO \square according to its measured coefficient of variation and re-running the model. The revised Methods section describes this procedure, and the resulting output range is shown as shaded bands in the new Fig. 1.

3. This is very much a top-down approach, allocating resources as a governing body (which is probably ok an early assessment). But in reality economic decisions will determine which match will be made, which will need an economic assessment/match too. I suggest adding this to the discussion. Similarly, now the target is an amount of rock supply, used to calculate ERW potential, because these numbers emerged from another study. For a first assessment this could be an ok approximation, but for emission reduction goals it should be reversed.

We thank the reviewer for highlighting these insightful points. We have added a paragraph at the end of the Discussion to address these limitations and suggest potential avenues for future research. These additions acknowledge the role of economic decision-making and outline how our model can be adapted, either via cost-based ranking or cost-threshold constraints, and applied in reverse to derive rock-supply requirements based of annual CDR targets.

4. While the positive impact to the soil (CO₂ uptake) is the main metric that is used and shown, potential negative effects are not mentioned (e.g. heavy metals). Considering that the framework is in place for the positive geochemical impact, I imagine it should not be very difficult to also calculate other (potentially negative) impacts? Although it would be nice to see such environmental analysis here, it would be sufficient if this is included in the discussion.

We thank the reviewer for highlighting the importance of potential negative impacts. Because our primary focus is on upstream rock supply, we have expanded our analysis of environmental impacts of expanded extraction sites by quantifying land-use change (Supporting Information Fig. S5) and discussing biodiversity loss and restoration opportunities in the discussion. We have also discussed the potential downstream soil risks, such as trace-metal mobilisation and pH shift in the Discussion section. These additions ensure that both positive and negative environmental effects are acknowledged and can be considered into future, more detailed multi-criteria deployment analyses.

5. Another suggestion which could be very interesting for investors and policy makers, is to overlay the extraction map with population density, which provides a first social impact assessment.

We thank the reviewer for this suggestion. In fact, we already undertook a preliminary social impact scoring in our previous work (Madankan & Renforth, 2023). We found visualising this overlay a bit challenging in providing meaningful insight. Instead, we used QGIS's zonal-statistics tool and the 1 × 1 km gridded population dataset (Reis et al., 2017) to quantify the resident population around each quarry. By applying logarithmic weights to populations within 1, 2.5, 5 and 10 km buffers, we derived a single "social score" for all active, inactive and potential new silicate-rock quarries, as well as for over 2,100 aggregate sites across the UK for comparison. These insights are detailed in Madankan & Renforth (2023), and to avoid repetition we have referred readers to that publication.

6. I suggest changing the scenario names. Although they are concise, there is no relation to the actual scenario content, hence it is more difficult for readers to follow.

We thank the reviewer for this suggestion. In response, we have added Table 1 to the manuscript to summarise each scenario's code name (retained from Kantzas et al. 2022 for consistency), a concise descriptive name, and their key differentiating parameters (target supply by 2070, per-quarry cap, and contributing quarry types). We believe this table greatly improves clarity and helps readers follow the scenarios without sacrificing the brevity of the original codes. Furthermore, we have included each scenario's descriptive name or key feature (depending on the context) in parentheses alongside its code where they appear in the text.

7. When introducing climate change in lines 24-29, the most recent citation is 2022. As research is moving fast, I suggest adding more recent literature. Moreover, the 1.5°C target is already passed, so mentioning the timeline of "climate" is essential.

We have revised the opening paragraph to incorporate up-to-date literature and an explicit timeline. It now cites the WMO confirmation that global mean temperature averaged ≈ 1.55 °C above pre-industrial levels in 2024 (WMO, 2025) and the IPCC AR6

Report statement that large-scale CDR is required to achieve net-negative CO₂ emissions (IPCC, 2023).

8. Line 48: It is mentioned that such a rock extraction increases in not unprecedented, but there is no mention of when or where this happened (or a reference).

We have clarified our statement by adding the specific historical example and citation: Such a substantial expansion within a relatively short timeframe, while not unprecedented (for example, a rise in the UK's crushed-rock production from ~ 25 Mt yr⁻¹ in 1945 to over 200 Mt yr⁻¹ by 1990, Kantzas et al., 2022), necessitates a robust plan for the spatial and temporal distribution of this upscaling to ensure a sustainable increase in rock supply.

9. Lines 58-61 say that there is a need for including environmental and social considerations, and that this study aims to address this need. But this paper does not include an environmental or social study? The problem statement/goals should therefore be rephrased.

We have revised the relevant sentences in the Introduction to clarify that the present work does not attempt a full environmental or social impact assessment. Instead, it incorporates a first-order proxy, extraction-density, as an initial indicator of potential extent of impacts under each up-scaling scenario

10. In my opinion it is not very clear from the start what the scenarios and their constraints are. This is somewhat scattered throughout the paper I suggest adding a table with the major parameters.

Thank you for this suggestion. We have added Table 1 to Results section, which collates the six scenarios, their codes and descriptive names, target rock-supply levels, per-quarry capacity caps, and contributing quarry types.

11. The result differences between the scenarios, presented in Table 1, are caused by two parameters: the production amount+quarry activation, and the cap per quarry. I suggest to present and discuss the results much more in function of these parameters. For example, it is unclear why the CDR rate is decreasing with increasing scenario freedoms (S1->S3). Is it caused by the lower cap per quarry?

We appreciate this insightful observation. In the revised manuscript, the Results / Cumulative rock supply section now explicitly links the observed differences between scenarios to the two key controls:

Supply target (scale–efficiency trade-off): Higher rock-supply targets force the model to include lower-yield quarry–cropland pairs, thereby reducing the mean CDR rate as we move from S1 to S3.

Per-quarry production cap: Tighter caps (e.g. 1 Mt yr⁻¹) spread extraction across more, and generally less optimal, quarries, whereas higher caps (e.g. 2 or 5 Mt yr⁻¹) allow concentration at the best-performing sites, helping to achieve CDR efficiency.

12. For figure 2, I suggest to join the legend components together, and explicitly mention what the area is for the rock spread amount (10*10km I assume).

We have fully updated the figure with a uniform, consistent colormap and used a single legend for the annual production of quarries in all scenarios. The grid size (10 × 10 km) is mentioned in the caption.

13. I suggest adding scenario S3.a to Figure 3 anyway, which will show the evolution over the scenarios even more clearly.

Figure 3 updated to include also scenario S3.a.

14. In Figure 4, the counties are used as cells for density calculation. However, using county-sized areas provides a very disturbed picture. For example: if there is one large quarry, for a large county it would show up lightly colored, while for a small county it would be darkly colored. I suggest to remake these maps with a uniform grid, e.g. 10*10km (see also lines 466-471).

To address this, we have generated a uniform 10 × 10 km grid map of extraction density (cumulative rock extracted per km²) for scenarios S3.a–S3.c and placed it in the Supporting Information (Fig. S6). We retain the county-level map in the main text (Fig. 4) both to maintain consistency with the other county-based metrics in that figure and because county boundaries align directly with local-authority planning jurisdictions, which can be useful for policy interpretation. A reference to the new grid-based maps and insights from them has been added to the “Results / Resource requirements, flow and logistics” section.

15. Figure 5: please indicate in the caption that these results are from Scenario Sc3.c.

Mentioned in the caption.

16. Figure 5b: Considering the freight growth since the 1960s, I'm not sure if assuming a stable amount of transport for the coming 50 years is realistic, just because it has been stable for last couple of years. Moreover, in lines 288-289 this leads to a false statement: it is a very rough assumption that freight amounts will stay stable. Drawing the conclusion that the additional required capacity is lower than the historical maximum is wrong.

We agree that assuming flat freight volumes over five decades is not realistic. Accordingly, we have added two cases based on projections from the NIC Future of Freight Demand report: a low-growth rate of 0.7 % yr⁻¹ and a high-growth rate of 1.1 % yr⁻¹. These annual rates are used to project total inland freight to 2070 and new graphs plotted along with additional demand from ERW. The updated figures 5.b and 5.c are now described in the Results section.

The statement “additional transport capacity required for ERW by 2070 will be lower than the historical maximum” also removed from the manuscript.

17. Line 295: Regarding the reference for the cost of port capacity increase: is there no example from the UK (or Europe) that is more representative?

Thank you for the suggestion. We conducted a thorough search of recent UK and European port projects, including London Gateway and several European ports but none publicly report either a capacity increase or corresponding CAPEX. The only dry-bulk port we found is the Gateway Pacific Terminal (USA), but its data dates back to 2011 and may no longer reflect current cost benchmarks.

18. Line 312: The discussion on the environmental and social impact ends quite abruptly, without having a proper discussion on environmental benefits vs downsides, social impact (equitable distribution...).

We have extended the Environmental & Social Impact section by further discussing the impacts such as land-use change projections and biodiversity risks, downstream soil risks, and suggesting benefit-sharing mechanisms for affected communities.

19. Line 330-331: Also consider permitting in the lead time.

We rephrased this part and added more accurate data: "According to AMPS (2023), crushed-rock planning applications take on average three years to determine, and the full development cycle, from initial exploration through to operation, can span 5-15 years."

20. Line 338 states that this study outlines routes for sustainable upscaling, but sustainability is not evaluated or taken into account in the model or matching. Especially marketing and popular media are very uncaringful in using "sustainability", I urge to be much more careful in science.

The word "Sustainable" was removed from the statement.

21. As far as I can find, when identifying potential new quarry areas only the surface area (from geological maps) are considered. However, thickness of the suitable formations is as important, has there been no reserve calculation for the UK?

We acknowledge that surface area alone does not guarantee reserve volume. In this study, the potential location for new quarries is suggested as a centroid of 10 x 10 km grid cells of suitable rock outcrops. To provide a first-order volumetric check, if we assume that within each 10 × 10 km grid cell (100 km²) there is at least 1 km² of extractable outcrop with at least 50 m thickness, then at a typical silicate-rock density of 2.7 t m⁻³, this corresponds to roughly 135 Mt of extractable rock per grid cell, sufficient for a single quarry operating at 2 Mt yr⁻¹ for over 50 years. We have added this into the Method section.

22. Line 427-428: Does the CDR potential calculation also account for the relative amount of suitable cropland per grid cell? Mention this explicitly if so.

No – the CDR potential is calculated per unit mass of rock (t CO₂ / t Rock) for each quarry–cropland grid pair. The area of suitable cropland in each grid cell is then explicitly incorporated during the spatio-temporal allocation step, where rock is allocated according to the available cropland area.

23. The publication by Kantzas et al. (2022), cited in lines 429 and 437, is not in the reference list, while it is a very important paper for reaching the current results.

The citations to "Kantzas et al. (2022)" have been corrected to "Kantzas et al. (2022)".

24. In Madankan & Renforth (2023), maps are published on the relative appropriateness of UK croplands for ERW (Figure 5 of that paper). It is very apparent that the most suitable areas from that paper are opposite to those of the current paper (e.g. Figure 2, partly also

Figure S4). Since this paper shares much data and methods with the Madankan & Renforth paper, why is this difference so apparent? I would at least mention this somehow in the methods or discussion.

First, the ERW suitability map in Madankan & Renforth (2023) includes both arable croplands and improved grasslands. In our study, we confined the analysis to arable land, where precise CDR estimates from Kantzas et al. (2022) and therefore omitted grasslands (which are predominantly located in the northwest). We have now clarified that in the Method section.

Second, Madankan & Renforth's map shows intrinsic CDR suitability per unit area. But, our Figure 2 depicts the volume of rock applied per grid cell, directly proportional to the area of arable land. Since arable cropland is most extensive in the southeast (consistent with Madankan & Renforth, 2023, Fig. S5.a), our rock-spread map highlights those regions. This is now clarified in the results section.

Furthermore, the ERW suitability map in Madankan & Renforth (2023) as a primary estimate has some limitation like sensitivity to parameters weighting as outlined in their supporting information (Fig. S2), this is why we used more precise CDR values from Kantzas et al (2022).

Response to Review

We thank the reviewer for the minor suggestions. Which we have addressed below.

1. In figure 1, only the first sub-figure received an (a) label; b, c and d are missing. **Corrected.**
2. The sentence in line 378 has, I believe, a grammar issue. **The sentence: Land use change can be of the key environmental impacts of expanded rock extraction corrected to: Land use change can be one of the key environmental impacts of expanded rock extraction (now in line 321).**
3. The reference in line 706 is written incorrectly. This is a report for the UK government, and although there are no authors mentioned, I think the reference should be: "Element Energy & UK Centre for Ecology & Hydrology, 2021. Greenhouse gas removal methods and their potential UK deployment. Report for the UK Department for Business, Energy and Industrial Strategy, 105 p. **Corrected, its now in reference No. 24, line 668.**